# Unveiling Multiple Descents in Unsupervised Autoencoders

**Kobi Rahimi**                                                               *kobirahimi@gmail.com*
*Faculty of Engineering, Bar-Ilan University,*
*Ramat-Gan 5290002, Israel.*

**Yehonathan Refael**                                                   *refaelkalim@mail.tau.ac.il*
*Faculty of Engineering, Tel Aviv University,*
*Tel Aviv 6997801, Israel.*

**Tom Tirer**                                                                   *tirer.tom@biu.ac.il*
*Faculty of Engineering, Bar-Ilan University,*
*Ramat-Gan 5290002, Israel.*

**Ofir Lindenbaum**                                                     *ofir.lindenbaum@biu.ac.il*
*Faculty of Engineering, Bar-Ilan University,*
*Ramat-Gan 5290002, Israel.*

**Reviewed on OpenReview:** *https://openreview.net/forum?id=FqfHDs6unx*

## Abstract

The phenomenon of double descent has challenged the traditional bias-variance trade-off in supervised learning but remains unexplored in unsupervised learning, with some studies arguing for its absence. In this study, we first demonstrate analytically that double descent does not occur in linear unsupervised autoencoders (AEs). In contrast, we show for the first time that both double and triple descent can be observed with nonlinear AEs across various data models and architectural designs. We examine the effects of partial sample and feature noise and highlight the critical role of bottleneck size in shaping the double descent curve. Through extensive experiments on both synthetic and real datasets, we uncover model-wise, epoch-wise, and sample-wise double descent across several data types and architectures. Our findings indicate that over-parameterized models not only improve reconstruction but also enhance performance in downstream tasks such as anomaly detection and domain adaptation, highlighting their practical value in complex real-world scenarios.

## 1 Introduction

In recent years, studies have shown that over-parameterized models outperform smaller models in generalization (Krizhevsky et al., 2012; He et al., 2016), challenging the traditional bias-variance tradeoff (Hastie et al., 2009). This is explained by the double descent phenomenon, extensively studied in supervised learning (Belkin et al., 2019; Nakkiran et al., 2021; Dar et al., 2021). However, double descent has yet to be demonstrated in a fully unsupervised setting, with one study arguing for the absence of the phenomenon in unsupervised autoencoders (AEs) (Lupidi et al., 2023).

In this study, we use AEs to explore double descent and its impact on key **unsupervised** tasks like domain adaptation, anomaly detection, and robustness to noisy data. AEs are widely used in unsupervised tasks such as denoising (Vincent et al., 2008; 2010), manifold learning (Wang et al., 2014; Duque et al., 2020), clustering (Song et al., 2013; Yang et al., 2019), anomaly detection (Sakurada & Yairi, 2014; Zhou & Paffenroth, 2017), feature selection (Han et al., 2018; Gong et al., 2022), domain adaptation (Deng et al., 2014; Yang et al., 2021), segmentation (Myronenko, 2019; Baur et al., 2021), and generative modeling (Kingma, 2013; Doersch, 2016), making them a prominent use case when studying double descent in unsupervised learning.

We start by analyzing linear AEs, whose lack of nonlinear activation functions allows for theoretical exploration. We prove and empirically confirm that double descent does not occur in linear unsupervised AEs. We then investigate nonlinear AEs and find that multiple descents emerge across different data models and architectures.

We provide extensive evidence showing the phenomenon in unsupervised nonlinear AEs trained on contaminated data, with "memorization" causing overfitting and performance degradation. However, we find that over-parameterized models can extract the true signal, leading to a second descent and improved performance.

Our experiments focus on synthetic and real-world datasets under various contamination scenarios, including noise, domain shifts, and outliers. Additionally, we identify model, epoch, and sample-wise double descent, highlighting the bottleneck size's role in shaping the double descent curve.

These findings provide critical insights into the dynamics of unsupervised learning and the interplay between model capacity, noise, and performance. The phenomenon is shown in Figure 1, illustrating the bias-variance tradeoff, the critical regime, and the second descent induced by over-parametrization of unsupervised AEs.

Our findings have important implications for key tasks in unsupervised learning. Specifically, we show that over-parameterized models adapt better to target domains when trained on source domains despite domain shifts. We also reveal non-monotonic performance in anomaly detection as model complexity increases, underscoring the practical relevance of our results, particularly when addressing outliers and domain shifts.

Our main **contributions** are summarized below:

(a) To the best of our knowledge, this is the first demonstration of model and epoch, and sample-wise double descent in a fully unsupervised setting. We analyze and contrast the behavior of linear and nonlinear AEs through analytical and empirical evaluations, highlighting their distinct dynamics.

(b) We analyze various factors influencing the presence of double descent, including sample and feature noise, domain shifts, anomalies, and architecture design (bottleneck size).

(c) We show that double descent in reconstruction loss of AE causes non-monotonic performance in real-world, downstream tasks like domain adaptation and anomaly detection.

## 2 Related Work

Most research on double descent has focused on supervised settings. Model-wise double descent was demonstrated in (Spigler et al., 2018), while (Li et al., 2020; Bartlett et al., 2020; Nakkiran et al., 2021; Gamba et al., 2022; Hastie et al., 2022) explore the impact of feature, label noise, and Signal-to-Noise ratio (SNR) on the double descent curve. (Nakkiran et al., 2021; Dubova, 2022; Gamba et al., 2022; Kausik et al., 2023; Sonthalia & Nadakuditi, 2023) demonstrated epoch-wise and sample-wise double descent, and multiple descents were discussed in (Adlam & Pennington, 2020; Liang et al., 2020; Chen et al., 2021).

While double descent is well-studied in supervised settings, its presence in unsupervised tasks is less understood. For example, (Lupidi et al., 2023) argued that model-wise double descent does not occur in unsupervised AEs, and (Gedon et al., 2022) found it absent in principal component analysis (PCA) (Shlens, 2014b). In contrast, supervised PCA tasks like principal component regression (PCR) (Massy, 1965) show evidence

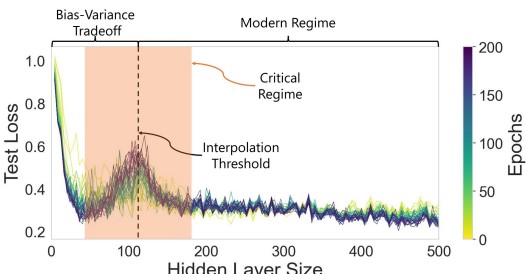

Figure 1: Demonstration of double descent phenomenon with unsupervised AEs for the "sample noise" scenario (Subsection 3.1). We present the test loss for varying epochs and hidden layer sizes.

of the phenomenon (Xu & Hsu, 2019; Teresa et al., 2022), highlighting key differences and challenges in observing double descent in unsupervised settings.

In this work, we study AEs' unsupervised objective of minimizing reconstruction error, $\|\Phi(\boldsymbol{x}) - \boldsymbol{x}\|_2^2$, where $\boldsymbol{x}$ is the input and $\Phi(\boldsymbol{x})$ is the AE's output. This learning framework, which has not previously shown double descent, falls outside the settings explored in prior studies. We find that linear AEs lack double descent, while nonlinear AEs show double and triple descent, underscoring the role of nonlinearity. Our analysis spans model, epoch, and sample levels, exploring the impact of bottleneck size, hidden layers, noise magnitude, and sample contamination. We also demonstrate double descent in real-world unsupervised scenarios like domain shifts, anomalies, and additive noise, with implications beyond reconstruction tasks.

## 3 Data Model

This section details the training data, testing data, and contamination models used to study double descent.

### 3.1 Subspace Data Model

We revisited the model used in (Lupidi et al., 2023), which argued that "double descent does not occur in self-supervised settings." We sampled $N$ i.i.d. Gaussian vectors of size $d$ ($\boldsymbol{z}_i \sim \mathcal{N}(0, \mathbb{I}_d)$), representing latent features and embedded them to a higher-dimensional space using $\boldsymbol{H}$ of size $D \times d$, ($D > d$, $\boldsymbol{H}_{ij} \sim \mathcal{N}(0,1)$). Our dataset differs from (Lupidi et al., 2023), and we explore four scenarios in our study:

**Sample Noise.** We study how the proportion of noisy training samples ($p$), affects the test loss curve. Unlike (Lupidi et al., 2023), which adds noise to all samples, we vary $p$ and use different SNR values (Appendix A, Table 1). This gives the following equation for the training samples:

$$\boldsymbol{x}_i^{D \times 1} = \begin{cases} \beta \boldsymbol{H} \boldsymbol{z}_i + \boldsymbol{\epsilon}_i, & \text{with probability } p, \\ \beta \boldsymbol{H} \boldsymbol{z}_i, & \text{with probability } 1 - p, \end{cases} \tag{1}$$

where $\boldsymbol{\epsilon}_i \sim \mathcal{N}(0, \mathbb{I}_D)$ represents noise added to samples with probability $p$ and $\beta$ controls the SNR.

**Feature Noise.** We study the impact of noisy training features on the test loss curve by selecting the same $\lfloor D \cdot p \rfloor$ features to be noisy across all samples. This simulates $\lfloor D \cdot p \rfloor$ unreliable or noisy measuring tools.

When noise is added to the data, we quantify the ratio between signal and noise power using the SNR. A high SNR indicates high-quality (clean) data, whereas a low SNR corresponds to lower-quality (noisy) data, where the noise dominates. We report SNR values in decibels, defined as SNR [dB] $= 20 \cdot \log_{10}\left(\frac{E[\|\beta \boldsymbol{H} \boldsymbol{z}\|_2]}{E[\|\boldsymbol{\epsilon}\|_2]}\right)$. To ensure the training data is sufficiently noisy, we consider cases where SNR $\leq 0$, meaning the noise power

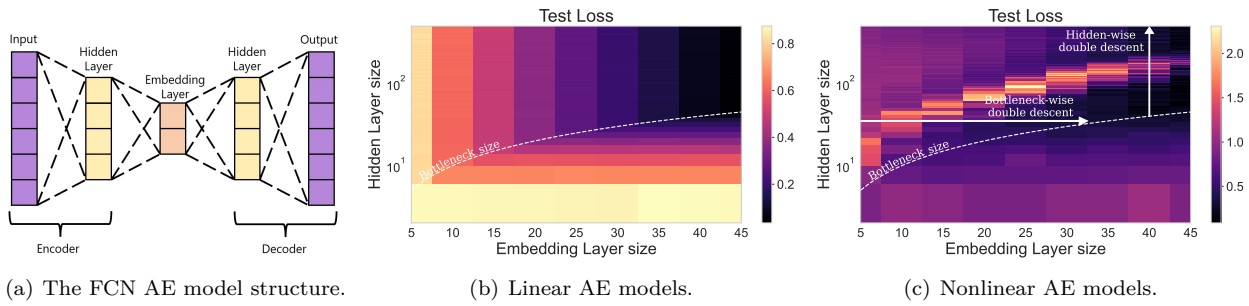

(a) The FCN AE model structure.    (b) Linear AE models.    (c) Nonlinear AE models.

Figure 2: (a): Illustration of our AE model. The embedding layer serves as a bottleneck when the hidden layers are larger. (b), (c): Test losses of linear and nonlinear AEs for different hidden and embedding sizes. Linear AEs do not exhibit double descent, whereas in nonlinear AEs, we clearly see the phenomenon when varying both the hidden layer and embedding layer size. Train losses are illustrated in Appendix D, Figure 18. AEs were trained on the subspace data model (Subsection 3.1) with 90% sample noise and SNR = -15 dB.

exceeds that of the signal. The SNR calculations for both sample and feature noise scenarios are presented in Appendix B, and a visualization of the data generation is illustrated in Appendix A, Figure 14.

**Domain Shift.** We analyze test loss behavior under domain shifts by projecting the test and train latent features using $\boldsymbol{H}$ and $\boldsymbol{H}''$ respectively, with a shift modeled as $\boldsymbol{H}'' = \boldsymbol{H} + s \cdot \boldsymbol{H}'$, where $\boldsymbol{H}'_{ij} \sim \mathcal{N}(0,1)$ adds perturbations, and $s > 0$ controls the shift,

$$\boldsymbol{x}_i = \begin{cases} \boldsymbol{H}\boldsymbol{z}_i, & \text{if } train, \\ \boldsymbol{H}''\boldsymbol{z}_i, & \text{if } test. \end{cases}$$

This data model is illustrated in Appendix A, Figure 15.

**Anomalies.** We examine how train-set anomalies affect the test loss curve. Normal samples are $\{\beta\boldsymbol{H}\boldsymbol{z}_i\}_{i=1}^{N}$, while anomalies follow $\mathcal{N}(0, \mathbb{I}_D)$. The Signal-to-Anomaly Ratio (SAR), controlled by $\beta$, sets their magnitude ratio. We replace $p \cdot 100\%$ of normal samples with anomalies. Illustration is shown in Appendix A, Figure 16.

We additionally provide results for a nonlinear subspace data model across all the aforementioned contamination scenarios (i.e., sample and feature noise, domain shift, and anomalies) in Appendix F.4.

In the following subsections (i.e., 3.2, 3.3, 3.4), we present real-world, high-dimensional datasets from diverse domains, trained on different architectures. This setup allows us to demonstrate the generalization of the double descent phenomenon across both datasets and model types. Due to their high dimensionality, these datasets are well-suited for dimensionality reduction via AEs, as also explored in prior work (Eraslan et al., 2019), (Yan et al., 2016), (Wang et al., 2016), (Meng et al., 2017).

### 3.2  Single-Cell RNA Data

We used single-cell RNA data from (Tran et al., 2020) to demonstrate our findings in a challenging, high-dimensional real-world setting. The dataset was chosen due to its inherent domain shifts, which include 5 distinct domains (biological batches) stemming from differences in laboratory conditions and measurement technologies. This makes it particularly suitable for testing our claims about double descent under real-world distribution shifts, which are common in biological and medical data (Haghverdi et al., 2018; Shaham et al., 2017). Each batch represents 15 different cell types, where each cell (sample) in this dataset contains over 15,000 genes (features), making it a high-dimensional dataset. Due to the real-world nature, we cannot control the shifts between the training (source) and test (target) datasets. This dataset is used for sample noise, feature noise, and domain shift experiments.

### 3.3  CelebA Data

The CelebA dataset (Liu et al., 2015) was selected to evaluate the impact of model complexity on unsupervised anomaly detection performance (Han et al., 2022). CelebA provides a rich, real-world setting with data attributes, making it particularly suitable for testing whether model complexity translates into non-monotonic trends in downstream tasks such as anomaly detection.

### 3.4  MNIST and CIFAR-10

We employed the MNIST (LeCun et al., 1998) and CIFAR-10 (Krizhevsky et al., 2009) datasets to evaluate our findings on standard image benchmarks, demonstrating double descent under sample noise, feature noise, and domain shift scenarios. These datasets also support reproducibility across different model architectures and data domains.

For more information about the nonlinear data model, single-cell RNA, celebA, MNIST, and CIFAR-10 datasets, refer to Appendix A.

## 4  Model Architecture

This section details the model architectures used to study double descent.

### 4.1 Fully-connected Neural Network (FCN)

We used FCNs (see architecture in Figure 2(a)) to study double and triple descent across all mentioned data models. FCNs are widely used with tabular data and have recently demonstrated state-of-the-art results in various tasks (Gorishniy et al., 2024a;b; Shenkar & Wolf, 2022; Rauf et al., 2024; Svirsky & Lindenbaum).

### 4.2 Convolutional Neural Network (CNN)

To illustrate the generality of our findings, we additionally present double descent phenomena using CNNs (architecture provided in Appendix A, Figure 17) trained on image datasets. A subset of the results is included in Subsection 6.1, while the majority, along with further experimental details, can be found in Appendix F.3. The results in the main paper for this architecture also include the training loss, whereas all other figures in the main paper, based on FCNs, show only test losses, with corresponding training losses provided in the Appendix. Across all experiments involving CNNs, the training loss consistently shows a monotonic decrease, similar to the other settings with FCNs.

## 5 No Double Descent in Linear AEs

This section presents theoretical analysis and empirical evidence showing that *linear* AEs do not display the double descent phenomenon. These results, along with our subsequent empirical findings regarding multiple descents in *nonlinear* AEs, highlight the crucial role of nonlinearity in shaping the test loss curve.

**Notation and setup.** Consider a general *linear* AE, $\Phi(\cdot; \boldsymbol{\theta}) : \mathbb{R}^D \to \mathbb{R}^D$, which consists of $L \geq 2$ layers and is parameterized by $\boldsymbol{\theta} \triangleq \left\{ \boldsymbol{W}_1^{d_1 \times D}, \dots, \boldsymbol{W}_{L-1}^{d_{L-1} \times d_{L-2}}, \boldsymbol{W}_L^{D \times d_{L-1}} \right\}$. Here, $\boldsymbol{W}_i$ represents the weights parameters associated with the $i$-th layer, for $i \in [L]$. For simplicity, we denote $\mathcal{D} = \{d_i \mid i \in [L]\}$ (including $d_L = d_0 = D$). Accordingly, we let $m = \min_{i \in [L]} d_i$ to be the *bottleneck* of the AE, namely, $\forall i \in [L], m \leq d_i$, and $m \leq D$. Formally, the model $\Phi$ is given by, $\Phi(\boldsymbol{x}; \boldsymbol{\theta}) = \boldsymbol{W}_L \boldsymbol{W}_{L-1} \cdots \boldsymbol{W}_2 \boldsymbol{W}_1 \boldsymbol{x}$.

Training the AE is based on empirical risk minimization:

$$\frac{1}{N} \min_{\boldsymbol{\theta}} \sum_{i=1}^{N} \mathcal{L}(\boldsymbol{\theta}; \boldsymbol{x}_i) = \min_{\boldsymbol{\theta}} \frac{1}{N} \sum_{i=1}^{N} \|\Phi(\boldsymbol{x}_i; \boldsymbol{\theta}) - \boldsymbol{x}_i\|_2^2.$$

According to the formulation of noisy training samples in (1), let $\mathcal{P}_{\boldsymbol{x}}$ denote the distribution from which the $\boldsymbol{x}_i$'s are drawn, i.e., $\boldsymbol{x}_i \sim \mathcal{P}_{\boldsymbol{x}}$. Similarly, let $\mathcal{P}_{\tilde{\boldsymbol{x}}}$ represent the distribution from which the **non**-noisy evaluation samples $\tilde{\boldsymbol{x}}_i$'s are drawn (see the bottom part of (1)). Then, for the trained $\Phi(\boldsymbol{x}; \boldsymbol{\theta}^{opt})$, where $\boldsymbol{\theta}^{opt} = \operatorname{argmin}_{\boldsymbol{\theta}} \frac{1}{N} \sum_{i=1}^{N} \mathcal{L}(\boldsymbol{\theta}; \boldsymbol{x}_i)$, we formalize the *generalization error* ($\mathcal{G}$) on the distribution of the **clean** data as:

$$\mathcal{G}(\mathcal{D}; \boldsymbol{\theta}^{opt}, \mathcal{P}_{\tilde{\boldsymbol{x}}}) = \mathbb{E}_{\mathcal{P}_{\tilde{\boldsymbol{x}}}} \left[ \|\Phi(\tilde{\boldsymbol{x}}; \boldsymbol{\theta}^{opt}) - \tilde{\boldsymbol{x}}\|_2^2 \right].$$

Note that the error depends on the AE architecture, which is characterized by the set of dimensions $\mathcal{D}$.

**Theoretical results.** The following theorem shows that double descent does not occur in linear AEs. We use the above notations, where a subscript ($a$ or $b$) distinguishes between different AEs.

**Assumption 5.1.** Let $\boldsymbol{X} = (\boldsymbol{x}_1, \boldsymbol{x}_2, \dots, \boldsymbol{x}_N) \in \mathbb{R}^{D \times N}$ be the training set, i.i.d. sampled as (1). Assume that the number of samples $N$ is large enough such that $\operatorname{rank}(\boldsymbol{X}) = D$ almost surely.

**Theorem 5.2.** *Let $\Phi_a(\cdot; \boldsymbol{\theta})$, $\Phi_b(\cdot; \boldsymbol{\theta}) : \mathbb{R}^D \to \mathbb{R}^D$ be two linear AEs, each with $L \geq 2$ layers and architectures defined by $\mathcal{D}_a$ and $\mathcal{D}_b$, with bottleneck sizes $m_a$ and $m_b$, where $m_a < m_b \leq D$. Let $\boldsymbol{\theta}_a^{opt}$ and $\boldsymbol{\theta}_b^{opt}$ be the optimal parameters under assumption 5.1. Let $d$ denote the dimension of the support of $\mathcal{P}_{\tilde{\boldsymbol{x}}}$ (the dimension of the span of the samples $\tilde{\boldsymbol{x}} \sim \mathcal{P}_{\tilde{\boldsymbol{x}}}$). Then, the following holds,*

$$\mathcal{G}\left(\mathcal{D}_b; \boldsymbol{\theta}_b^{opt}, \mathcal{P}_{\tilde{\boldsymbol{x}}}\right) \leq \mathcal{G}\left(\mathcal{D}_a; \boldsymbol{\theta}_a^{opt}, \mathcal{P}_{\tilde{\boldsymbol{x}}}\right),$$

*with equality if: (1) $m_a = m_b$, even if $\mathcal{D}_a \neq \mathcal{D}_b$, or (2) $m_a, m_b \geq D$.*

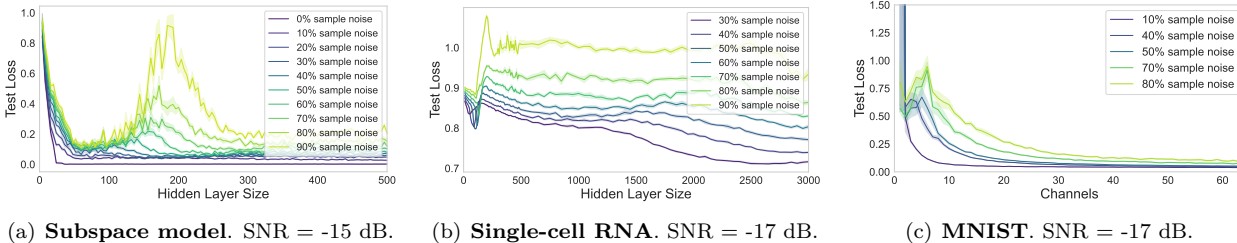

(a) **Subspace model**. SNR = -15 dB.  (b) **Single-cell RNA**. SNR = -17 dB.  (c) **MNIST**. SNR = -17 dB.

Figure 3: Model-wise double and triple descents for varying **sample noise**. Train losses are depicted in Appendix D, Figure 19.

Recall that the models are trained using noisy data sampled from $\mathcal{P}_{\boldsymbol{x}}$ and tested on clean data drawn from $\mathcal{P}_{\tilde{\boldsymbol{x}}}$ to see if there is noise memorization (high test loss) or signal learning (low test loss).

The proof of Theorem 5.2 can be found in Appendix C. This result shows that the $\mathcal{G}$ (test loss) decreases as the bottleneck size $m$ increases. Interestingly, it implies that the $\mathcal{G}$ remains unchanged if the bottleneck dimension is preserved, even if the size of other layers increases. Since the proof does not depend on the similarity between $\mathcal{P}_{\boldsymbol{x}}$ and $\mathcal{P}_{\tilde{\boldsymbol{x}}}$, the theorem holds for scenarios where $\mathcal{P}_{\boldsymbol{x}} \neq \mathcal{P}_{\tilde{\boldsymbol{x}}}$, including the cases of domain shifts and anomalies discussed in Subsection 3.1.

**Empirical results.** To validate our theory, we used the data model in Subsection 3.1 and an FCN AE without nonlinear activations (Figure 2(a)). Training linear AEs of varying sizes (Figure 2(b)) shows that test loss decreases as bottleneck size increases and is unaffected by non-bottleneck layer size. These results align with our theorem, explaining the absence of double descent in linear AEs.

# 6 Double Descent in Nonlinear AEs

We demonstrate the double descent phenomenon using FCN undercomplete AEs (Figure 2(a)), minimizing mean squared error (MSE) between inputs and outputs. Results are based on 5 to 15 random seeds, with implementation details in Appendix A. We trained models on contaminated datasets, drawn from $\mathcal{P}_{\boldsymbol{x}}$ and tested on clean data sampled from $\mathcal{P}_{\tilde{\boldsymbol{x}}}$ to isolate the effect on the learned model, showing noise memorization (high test loss) versus signal learning (low test loss) and observed model, epoch, and sample-wise multiple descents. Unlike standard analyses of PCA (Josse & Husson, 2012; Minka, 2000), where both training and test data are typically noisy and the goal is to recover the underlying signal by attenuating noise-dominated components, our setup deliberately uses clean test data to isolate the effect of noise memorization in the learned model. Train loss results of the FCNs are detailed in Appendix D, with additional findings on nonlinear synthetic datasets in Appendix F.4. We also present double descent results for CNN AE models (Appendix A, Figure 17) in Subsection 6.1, Figures 3(c), 4(c), 5(b) and in Appendix F.3.

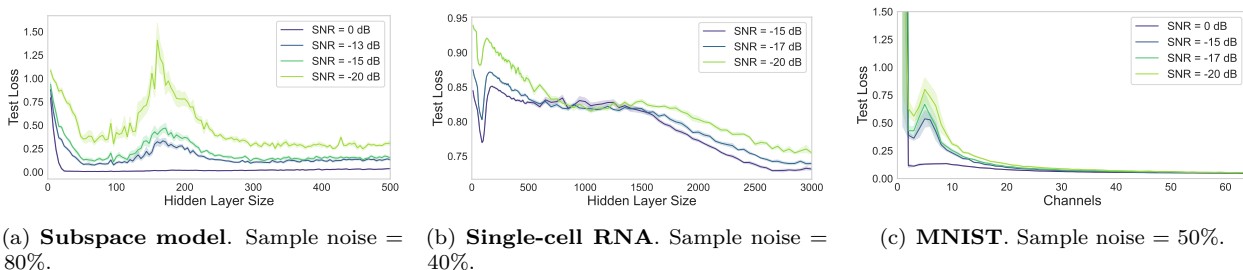

(a) **Subspace model**. Sample noise = 80%.  (b) **Single-cell RNA**. Sample noise = 40%.  (c) **MNIST**. Sample noise = 50%.

Figure 4: The effect of SNR (**sample noise** case) on the test loss curve. Train losses are illustrated in Appendix D, Figure 20.

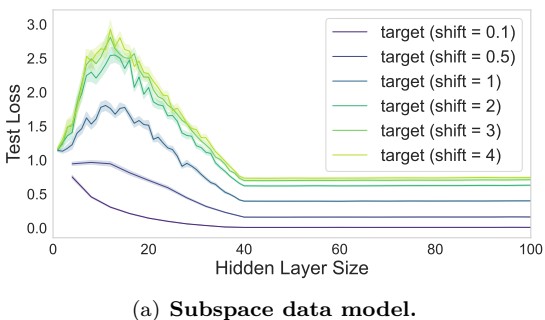 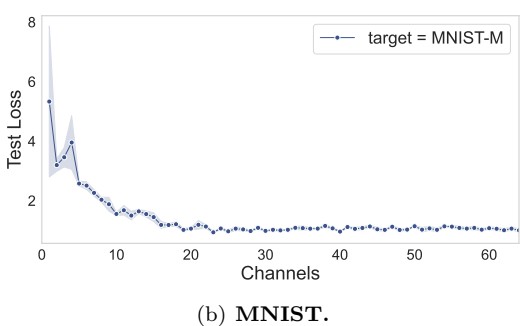

(a) **Subspace data model.**  (b) **MNIST.**

Figure 5: Model-wise non-monotonic behavior and double descent for varying **domain shifts**. Appendix D, Figure 21 shows the train loss behavior.

## 6.1 Model-Wise Double Descent

This section analyzes test loss with increasing model sizes, breaking down the *double descent* phenomenon into *hidden-wise* and *bottleneck-wise* variations and showing their impact on test loss. We find that various contaminations (Section 3) manipulate the interpolation threshold location and value. Double descent is more common with severe contaminations, as models in the critical regime interpolate noise instead of learning the signal, leading to higher test loss.

Figure 2(c) illustrates double descent in nonlinear AEs, concluding that nonlinear activations cause this phenomenon in unsupervised AEs. We define *bottleneck-wise* and *hidden-wise* double descent to distinguish between various model sizes and highlight the significance of our architectural choices. FCN undercomplete AEs ($m < D = 50$) have a bottleneck set by the smallest layer, with the critical regime and second descent (improving reconstruction) occurring when hidden layers exceed this size (above the white dashed line). Increase in bottleneck shifts the critical regime. Hidden layers above the dashed line set the embedding layer as the bottleneck, while smaller hidden layers become the bottleneck (Figure 2(a)). FCN overcomplete AEs ($m \geq D$) trivially learn the identity function and excluded from this study (see Appendix F.2, Figure 34).

**Sample and feature noise.** Figure 3(a) (FCN) shows that increasing sample noise raises test loss, as higher noise levels dominate training and degrade performance. Higher noise levels also shape the double descent curve by shifting the critical regime to larger models, which are needed to memorize more noise. At low noise levels such as 0–20% sample noise, double descent is absent due to insufficient noise in the training data, making models in the critical regime learn the signal and preventing a test loss increase. This underscores noise as a key factor in the phenomenon. Figure 3(b) (FCN) demonstrates triple descent in single-cell RNA data, with similar test loss behavior across both interpolation phases. Appendices E, F.3, F.4 provide additional evidence of double descent with feature noise. Figure 3(c) presents double descent results for CNNs trained on the MNIST dataset. Additional results of CNNs trained on MNIST and CIFAR-10 datasets and further details regarding these experiments are presented in Appendix F.3.

We also observed *final ascent*, where test loss increases after double descent, mirroring patterns seen in supervised learning (Xue et al., 2022). This is demonstrated for single-cell RNA data with 0–20% sample noise (Appendix F.5, Figure 56(b)). To demonstrate the generality of the phenomenon, we also report double and triple descent under Laplacian noise and in sparse AEs (Appendix F.6).

**SNR.** Figure 4 shows that SNR plays a key role in shaping test loss and revealing double descent. Negative SNR lets noise dominate the training set, causing memorization in the critical regime and exposing double descent. As SNR increases, noise influence diminishes, allowing better signal learning and lowering test loss. At SNR = 0 dB (Figures 4(a), 4(c)), double descent is absent or barely recognizable because noise is not dominant enough to trigger memorization.

**Domain shift.** Our experiments reveal non-monotonic behavior in test loss under domain shifts for the subspace data model using FCNs (Figure 5(a)). As the shift increases, test loss rises, but low shifts (0.1, 0.5) show no non-monotonicity due to strong source-target alignment, allowing interpolating models to perform well. Over-parameterized models further improve target reconstruction by reducing test loss. We also present

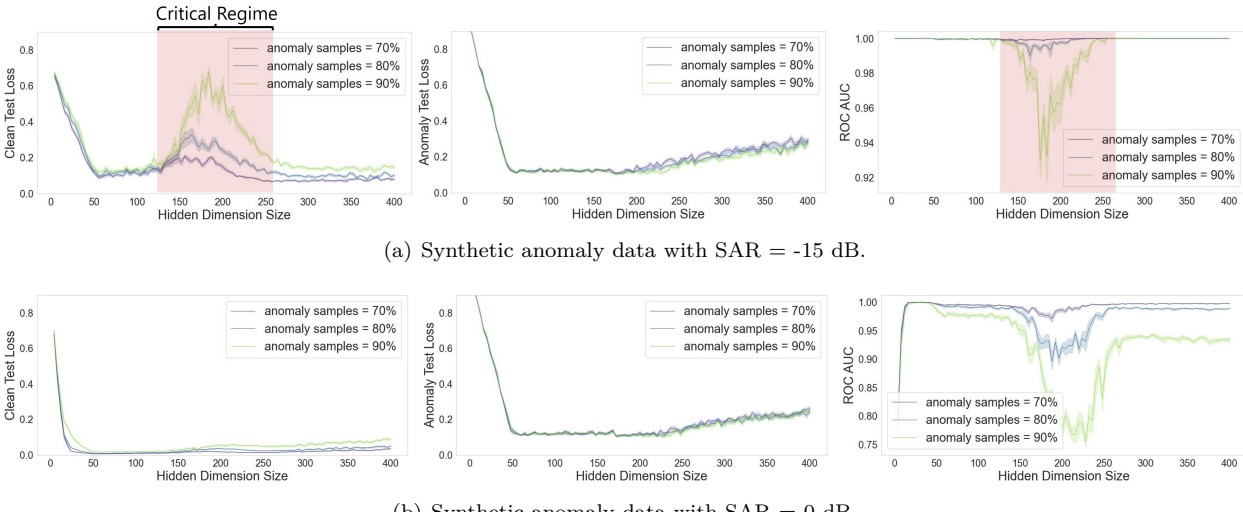

(a) Synthetic anomaly data with SAR = -15 dB.

(b) Synthetic anomaly data with SAR = 0 dB.

Figure 6: **Left column:** test loss of the clean (normal) samples. A double descent pattern emerges for low SARs and high anomaly presence in the training data. **Middle column:** test loss computed only over the anomalous samples. **Right column:** Non-monotonic behavior of the ROC-AUC. The level of SAR, as illustrated in (a) and (b), influences the occurrence of double descent in the test curve.

double descent behavior when CNNs are trained on MNIST and tested on the MNIST-M dataset in Figure 5(b). Subsection 7.1, Figure 12, demonstrates double and triple descent in single-cell RNA data under real domain shift, reinforcing the phenomenon's presence with real-world evidence.

**Anomalies.** For the first time, we show that anomalies can cause double descent in test loss and non-monotonicity in anomaly detection (Cheng et al., 2021). Using data from 3.1, we analyze test loss and detection quality via receiver operating characteristic area under the curve (ROC-AUC) (Hanley & McNeil, 1982; Fawcett, 2006), which distinguishes clean (low reconstruction error) from anomalous samples (high error) (Malhotra et al., 2016; Borghesi et al., 2019; Lindenbaum et al., 2024). In the critical regime (Figure 6(a)), anomaly memorization increases test loss for clean samples and reduces ROC-AUC. Larger models exhibit secondary descent, especially with low SAR and many anomalies, improving anomaly detection to match under-parameterized models while lowering test loss (Lerman & Maunu, 2018b;a; Han et al., 2022; Lindenbaum et al., 2024). Figure 6(b) shows no double descent at high SAR, similar to previous results where contamination is not significant, yet critical regime models still perform the worst. Additional real-world insights are in Subsection 7.2.In this case we keep $\mathcal{P}_{\boldsymbol{x}}$ and $\mathcal{P}_{\tilde{\boldsymbol{x}}}$ the same to evaluate model performance on both clean and anomalous data to generate the ROC curve and compute the AUC, providing a meaningful metric for assessing anomaly detection and demonstrating the relevance of our findings to downstream tasks.

In conclusion, factors such as model nonlinearity, bottleneck size, and severe contaminations, like high noise levels, large domain shifts, many anomalies, and low SNR, affect double descent and increase test loss, occasionally shifting the critical regime.

## 6.2 Epoch-Wise Double Descent

Inspired by the surprising results in (Nakkiran et al., 2021), which demonstrate epoch-wise double descent in a supervised setting, we study the existence of this phenomenon in our unsupervised framework. Figures 7 and 29 (Appendix E) show how noisy samples (and features) affect the test loss, respectively. Higher noise or lower SNR increases test loss, as shown in Figure 8 and Appendix E, Figure 30. The phenomenon also occurs with domain shifts (Figure 9), where stronger shifts lead to higher test loss, shown in Figure 9(a).

### 6.3 Sample-Wise Double Descent

This section examines how the number of training samples affects the test loss. Model complexity and sample size determine whether a model is over- or under-parameterized, shifting the interpolation threshold, as shown in Figure 10. This shift can sometimes make larger training sets lead to worse performance than smaller ones, a phenomenon also observed in supervised settings by (Nakkiran et al., 2021).

We analyze the effect of increasing training samples while keeping model size fixed and observe non-monotonic trends in the test loss curve (Figures 11(b), 11(c), and Appendix E Figure 31). In some cases, this leads to double descent, as shown in Figure 11(a). Appendix D and Figure 26 present the training losses, along with additional results using more complex data models found in Appendices F.3 and F.4. These results confirm the effects of noise, SNR, and domain shift, which are consistent with the findings in Subsections 6.1 and 6.2.

## 7 Real World Applications

This section highlights the application of our findings to critical tasks in unsupervised learning, such as domain adaptation and anomaly detection, focusing on the importance of model size selection in AEs rather than competing with state-of-the-art methods for these tasks.

### 7.1 Domain Adaptation

Domain shifts are common in machine learning, where differences between training and testing distributions can degrade performance on unseen data. Various domain adaptation methods (Peng et al., 2019; Chang et al., 2019; Zhou et al., 2022; Rozner et al., 2023; Yampolsky et al., 2023) aim to minimize this shift. In biology, this challenge, known as the "batch effect", arises when integrating datasets collected under different conditions (Tran et al., 2020).

This section explores the relationship between model size and its ability to address distribution shifts in real-world single-cell RNA data (Subsection 3.2). We demonstrate the benefits of over-parameterized models in unsupervised tasks under real domain shifts, revealing multiple descents. Related findings in supervised learning were shown by (Tripuraneni et al., 2021) and (Kausik et al., 2023) under different assumptions. Appendix F.1, Figure 32(b) visualizes source and target datasets using UMAP embeddings (McInnes et al., 2018). Figures 12(a), 12(b) show test and train losses for models trained on source and tested on target datasets, where testing on the 'Wang' dataset revealed triple descent.

Since we are working with a real-world dataset, we do not have control over $\mathcal{P}_x$ and $\mathcal{P}_{\tilde{x}}$. In particular, we cannot control the distributional shift between the training and test sets. We used KL-divergence (KLD) (Shlens, 2014a) to quantify the distribution shift between the source and target datasets, as shown in the test loss legend in Figure 12(a). Higher KLD values correspond to larger shifts and increased test loss, consistent with the simulated results in Figure 5(a).

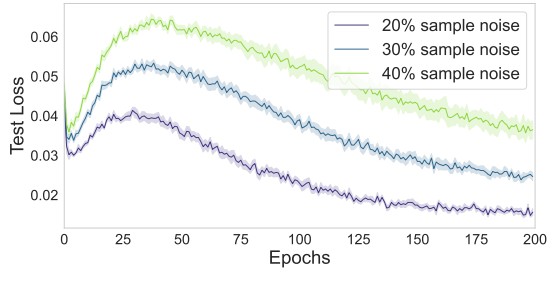

(a) **Subspace data model**. SNR = -2 dB.

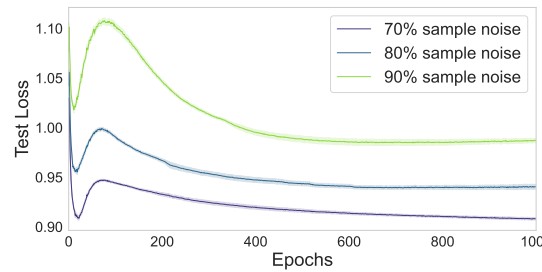

(b) **Single-cell RNA data**. SNR = -17 dB.

Figure 7: Epoch-wise double descent for different number of **noisy samples**. Train losses are depicted in Appendix D, Figure 22.

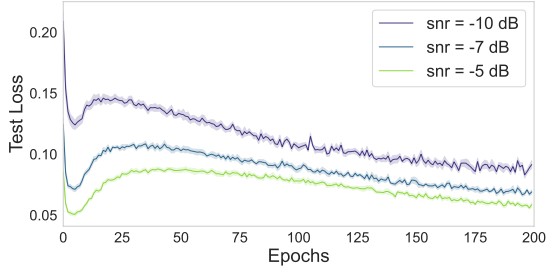 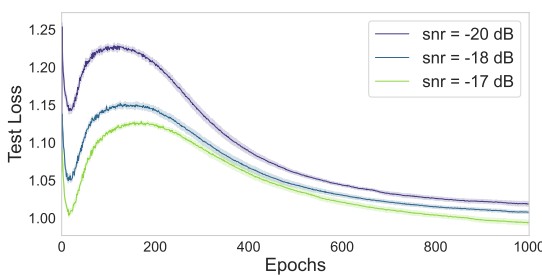

(a) **Subspace data model**. Sample noise = 40%.

(b) **Single-cell RNA data**. Sample noise = 90%.

Figure 8: Epoch-wise double descent (**sample noise**) influenced by the SNR. Train losses are exhibited in Appendix D, Figure 23.

To assess domain adaptation, we analyzed the AEs embeddings learned in the bottleneck layer using a $k = 10$ nearest neighbors domain adaptation test (KNN-DAT) (Schilling, 1986), Section 3, measuring domain mixing. KNN-DAT of 1 indicates complete separation and lower values signify better mixing. Improved mixing reflects embeddings where target samples are more aligned with source samples.

Figures 12(c), 12(d), 12(e) show UMAP representations based on AE embeddings. Under-parameterized models achieve better KNN-DAT than critical-regime models but at the cost of poor source data learning (high train loss). Critical-regime models focus on domain-specific features, leading to higher KNN-DAT. Over-parameterized models perform best, achieving a KNN-DAT of 0.75, reduced test loss, and improved target data reconstruction. This highlights their effectiveness, as *over-parameterized models facilitate the transition between source and target datasets, serving as a viable domain adaptation strategy*. Additional results for the subspace data model are in Appendix F.1.

## 7.2 Anomaly Detection

Unsupervised anomaly detection is vital in machine learning, with applications across various fields. AEs are widely used for this task (Chandola et al., 2009; Chen et al., 2018; Rozner et al., 2024; Lindenbaum et al., 2024). We trained AEs on both normal and anomaly data, detecting anomalies via reconstruction loss as detailed in Subsection 6.1. As noted in Subsection 6.1 (the 'Anomalies' section), we align $\mathcal{P}_{\boldsymbol{x}}$ and $\mathcal{P}_{\tilde{\boldsymbol{x}}}$ to assess the model's anomaly detection capabilities in the context of downstream tasks.

We examined how model size impacts anomaly detection using the CelebA dataset (Subsection 3.3). As shown in (Han et al., 2022), small models outperform larger ones, achieving the highest ROC-AUC (Figure 13). With uncontrolled positive SAR, we observed no double descent in test loss but identified non-monotonic

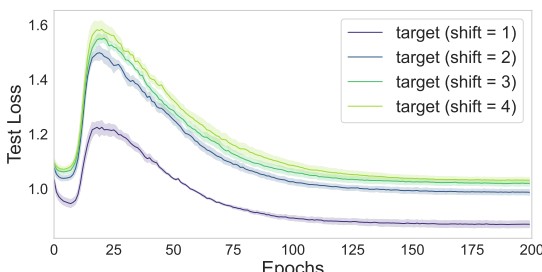 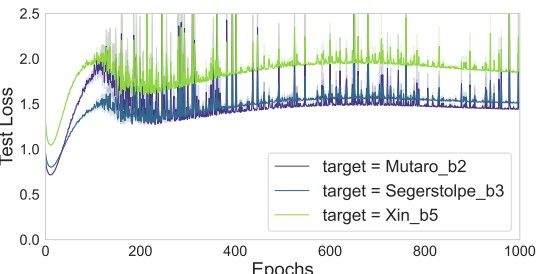

(a) **Subspace data model** with 5% noisy samples and SNR = 20 dB.

(b) **Single-cell RNA data**. The 'Wang' batch was excluded due to noisy results.

Figure 9: Epoch-wise double descent influenced by the **domain shift**. In (a) we introduce some noise to emphasize the double descent curve. Train losses shown in Appendix D, Figure 24.

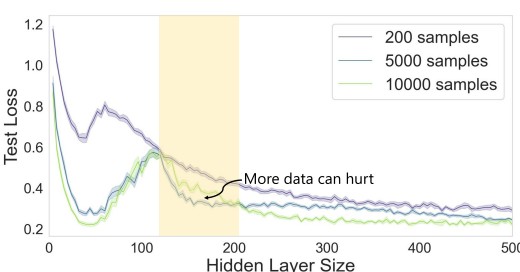

Figure 10: In the yellow interval, more data hurts performance as using 10000 samples worsens test loss compared to 5000. We used the subspace data model with sample noise of 70% and SNR of -15 dB. Train loss results are detailed in Figure 25, Appendix D.

ROC-AUC behavior. Thus, *intermediate models should be avoided, as their anomaly detection performance lags behind under and over-parameterized models.*

# 8 Discussion

In this section, we identify and highlight several fundamental distinctions between double descent behavior in supervised learning and the unsupervised setting explored in this work.

**Noise structure and data model.** In supervised learning, the input $x$ is typically clean, while noise is introduced in the labels $y$. In contrast, our AEs operate under an unsupervised reconstruction objective where the input serves as both source and target. Consequently, any corruption affects both sides of the learning signal (see last part of Section 2 and Section 3), inducing distinct generalization dynamics not observed in supervised settings.

**Presence of double descent in linear models.** Double descent has been extensively observed in supervised linear models such as linear regression (Belkin et al., 2019; Bartlett et al., 2020). However, our theoretical and empirical results demonstrate that linear unsupervised AEs do not exhibit double descent. This highlights a structural divergence between supervised and unsupervised learning, where nonlinearity becomes essential for the emergence of multiple descent phases.

**Training objective vs. learning dynamics.** While both supervised models and AEs often minimize the MSE, the training dynamics differ significantly. In supervised learning, the test distribution generally matches the training distribution. In contrast, our approach involves training on noisy inputs and evaluating on clean data, creating a distribution shift and affecting how generalization and overfitting emerge during training.

**Interpretation of over-parameterization.** In supervised tasks, interpolation is typically tied to the total number of parameters. Over-parameterization can be achieved by redistributing capacity across layers without necessarily increasing model size. However, in our setup, achieving interpolation requires joint scaling of the hidden and bottleneck layers (see Figure 2(c) and Subsection 6.1).

# 9 Conclusions

Our study comprehensively studied double descent in unsupervised learning using AEs. Analytically and empirically, we showed that linear AEs lack double descent. In contrast, nonlinear AEs exhibit multiple

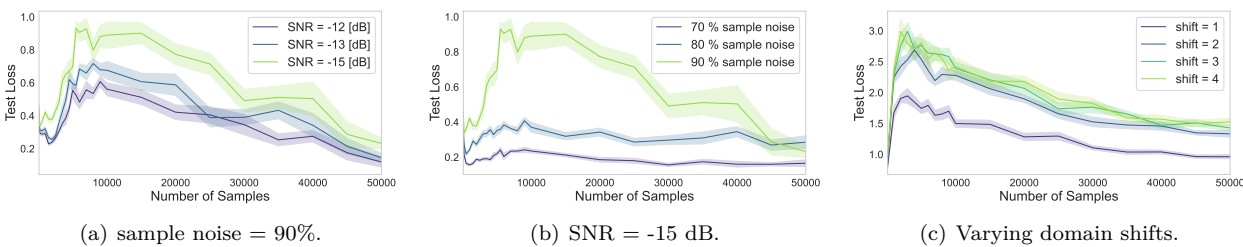

(a) sample noise = 90%.       (b) SNR = -15 dB.       (c) Varying domain shifts.

Figure 11: **Sample-wise** non-monotonicity and double descent for the **subspace data model**. Feature noise scenario results are presented in Appendix E, Figure 31 and the training losses in Appendix D, Figure 26.

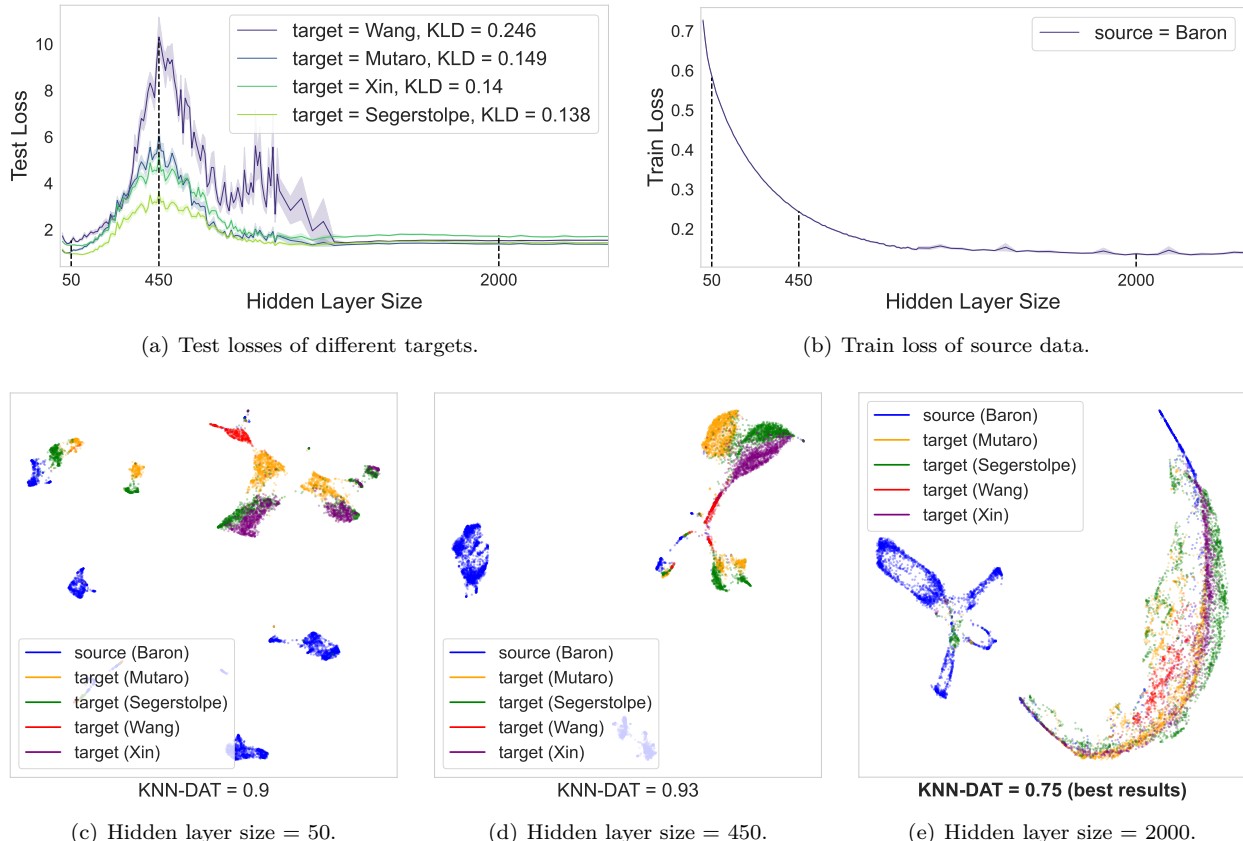

Figure 12: Double descent in real-world domain shift. (a), (b): Test and train losses utilizing the **single-cell RNA** dataset. Models are trained on source data and evaluated on target data. (c), (d), (e): UMAP of vectors extracted from the encoder's output and KNN-DAT results for different model sizes. As can be seen, Over-parameterized models ease adaptation between source and target datasets.

descents and non-monotonic behaviors across model-wise, epoch-wise, and sample-wise levels for various data models and architectural designs. We analyzed the effects of sample and feature noise and emphasized the importance of bottleneck size in shaping double descent. Our findings also show that over-parameterized models enhance both reconstruction and downstream tasks like anomaly detection and domain adaptation, underscoring their real-world relevance. Although our experiments were conducted on datasets of moderate size due to computational constraints, we believe the scope and depth of the evaluation offer a strong and conclusive foundation. This work raises broader implications for the use of model capacity in unsupervised settings. It suggests that double descent is not unique to supervised learning and must be considered when designing AEs for real-world applications. Several open research questions emerge from our study, including the need for a deeper theoretical understanding of double descent in unsupervised models, how different

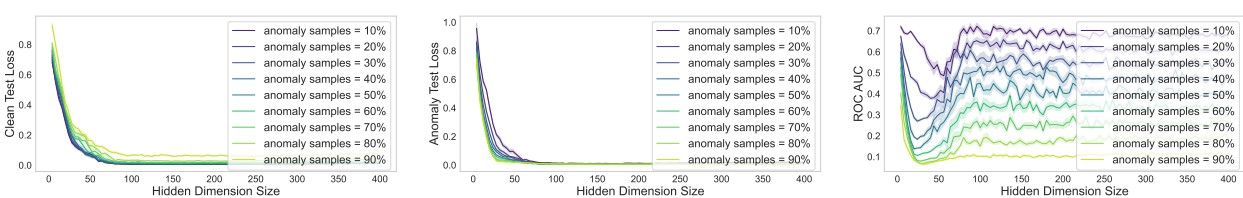

Figure 13: **Left, middle:** test loss of clean and anomaly data. **Right:** non-monotonic behavior of the ROC-AUC for the celebA dataset.

levels of nonlinearity influence the phenomenon, and the role of architecture-specific factors in shaping generalization behavior. We hope these findings contribute to the growing understanding of learning dynamics in unsupervised models and inspire future research into both theoretical and practical aspects of model generalization in unsupervised learning frameworks.

## Acknowledgment

TT was supported by ISF grant No. 1940/23 and MOST grant No. 0007091. OL was supported by MOST grant No. 0007341.

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

# A Implementation Details

This section provides complete implementation details for all experiments conducted in the paper. Illustrations of the subspace data model generation introduced in Subsection 3.1 for the scenarios of sample and feature noise, domain shift, and anomalies are displayed in Figures 14, 15, and 16 respectively.

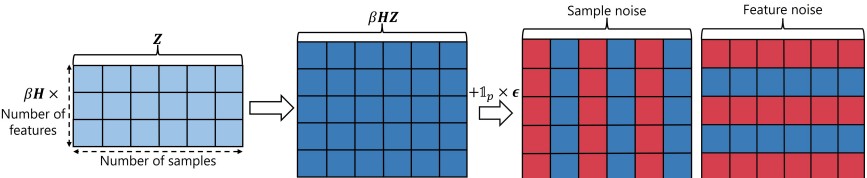

Figure 14: Train samples generation for the scenarios of sample and feature noise with $p = 0.5$. The first (leftmost) matrix depicts the latent vectors $\boldsymbol{Z}$. The second matrix illustrates the latent vectors being projected into a higher dimensional space, and the rightmost matrices contain clean (blue) and noisy (red) samples / features respectively. The test samples remain clean.

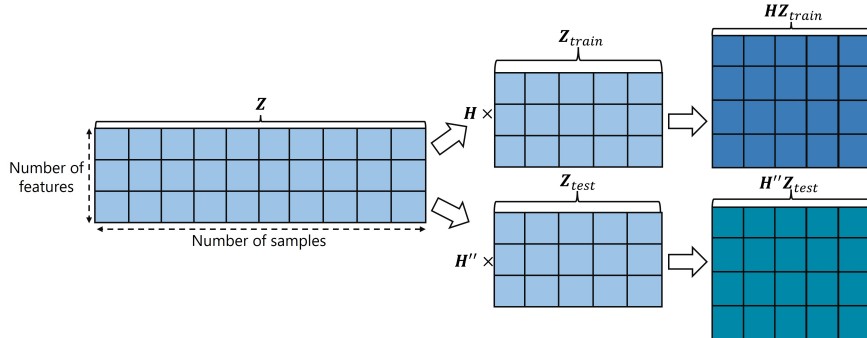

Figure 15: Train and test samples generation for the scenario of domain shift. The matrix on the left depicts the latent vectors $\boldsymbol{Z}$ and the two middle matrices represent the separation to source ($\boldsymbol{Z}_{train}$) and target ($\boldsymbol{Z}_{test}$). The two rightmost matrices illustrate the latent vectors of the train and test data being projected into a higher dimensional space with different matrices ($\boldsymbol{H}, \boldsymbol{H}''$), resulting in a domain shift.

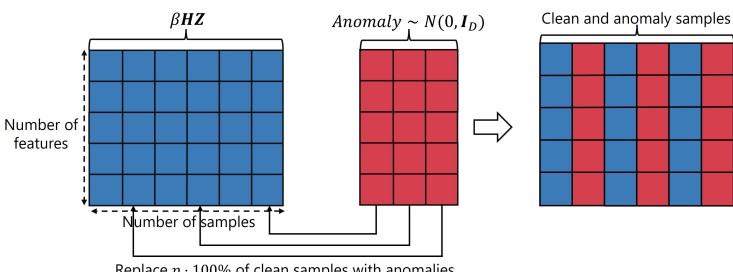

Figure 16: Train samples generation for the case of anomalies and $p = 0.5$. The matrix on the left depicts the clean data, the middle matrix represents the anomaly samples, and the rightmost matrix contains both clean (blue) and outlier (red) samples. Test samples remain clean.

**Parameters.** Table 1 details the hyper-parameters and other variables for the training process with the subspace data model, nonlinear subspace model, (Appendix F.4), single-cell RNA, CelebA, MNIST, and CIFAR-10 (Appendix F.3) datasets. The training optimizer utilized was Adam (Kingma & Ba, 2014), and the loss function for reconstruction is the mean squared error, which is mentioned in this Section.

Table 1: Parameters and hyper-parameters

| PARAMETERS | LINEAR/ NONLINEAR SUBSPACE | RNA | CELEBA | MNIST/ CIFAR-10 |
|---|---|---|---|---|
| MODEL | FCN | FCN | FCN | CNN |
| LEARNING RATE | 0.001 | 0.001 | 0.001 | 0.001 |
| OPTIMIZER | ADAM | ADAM | ADAM | ADAM |
| EPOCHS | 200 | 1000 | 200 | 1000 |
| BATCH SIZE | 10 | 128 | 10 | 128 |
| DATA'S LATENT FEATURES SIZE ($d$) | 20 | - | - | - |
| NUMBER OF FEATURES ($D$) | 50 | 1000 | 40 | 784/ 3072 |
| TRAIN DATASET SIZE | 5000 | 5000 | 3000 | 5000 |
| SNR/ SAR [$dB$] | -20, -18, -17, -15, -10, -7, -5, -2, 0, 2 | | | |
| CONTAMINATION PERCENTAGE ($p$) | 0, 0.1, 0.2,...,1 | | | |
| DOMAIN SHIFT SCALE ($s$) | 1, 2, 3, 4 | - | - | - |
| EMBEDDING LAYER SIZE | 25, 30, 45 | 20, 100, 300 | 25 | 10, 30, 50, 500 |
| HIDDEN LAYER SIZE | 4 - 500 | 10 - 3000 | 4-400 | - |
| CHANNELS | - | - | - | 1-64 |

**Data.** For the **subspace data model**, **nonlinear subspace data model**, **MNIST**, and **CIFAR-10** datasets, we generate 5000 samples for training and 10000 for testing across all scenarios (sample noise, feature noise, domain shift, and anomaly detection). Regarding the **single-cell RNA** data, we have focused on dataset number 4 from (Tran et al., 2020), which includes 5 distinct domains (biological batches) named 'Baron', 'Mutaro,' 'Segerstolpe,' 'Wang,' and 'Xin', each representing 15 different cell types. Each cell (sample) in this dataset contains over 15000 genes (features). To facilitate the training of deep models while preserving the domain shift, we have retained the top 1000 prominent features. We utilize the 'Baron' biological batch as our source data for the scenario of domain shift, comprising 5000 training samples, while the target batches are 'Mutaro' (2122 samples), 'Segerstople' (2127 samples), 'Wang' (457 samples), and 'Xin' (1492 samples). As for the sample and feature noise scenarios, we use the 'Baron' domain for both sample and feature noise scenarios due to its largest sample size (8569). We allocate 5000 samples for training and introduce noise to specific samples and features, as described in subsection 3.1. The calculations of the SNR for both sample and feature noise cases are provided in Section B. The reserved 3569 samples are for testing. Please be aware that all the domains in this dataset are inherently noisy, reflecting their real-world nature. Therefore, even when no additional noise is applied ($p = 0$), the data remains noisy. This may account for why the test loss does not decrease monotonically as the model size increases for cases with low noise levels, as shown in Appendix F.5, Figure 56(b).

For the **celebA** dataset, including over 200K samples and 4547 anomalies, each characterized by 40 binary attributes, we sub-sample 3000 clean samples and replace $\lfloor 3000 \cdot p \rfloor$ of them with anomalies. This ensures that $\sim p \cdot 100\%$ of the data is contaminated with anomalies. Due to the limited availability of anomaly data (4547 samples), the test set includes $\lfloor (1 - p) \cdot 4547 \rfloor$ anomalies along with an equal number of clean samples.

**Models.** All experiments, including the subspace data model, nonlinear subspace data model, single-cell RNA, and celebA datasets are conducted using the same FCN AE architecture. To facilitate the exploration of double descent in both embedding layer size and hidden layer size, we employ a simplified model mentioned in (Lupidi et al., 2023) consisting of a single hidden layer for both the encoder and decoder, as depicted in Figure 2(a). We also utilize a CNN AE architecture consisting of three convolution layers in the encoder part, followed by a bottleneck layer, and then a decoder part consisting of three deconvolution layers trained on the MNIST and CIFAR-10 datasets as illustrated in Figure 17 (results for the CNN AE are reported in Appendix F.3).

We work with undercomplete AEs to encourage the acquisition of a meaningful embedding in the latent space and prevent the model from learning the identity function. The size of these models is determined by the sizes of the hidden layers (for FCN), the number of channels (for CNN), and the bottleneck layer, while the width of these models remains constant.

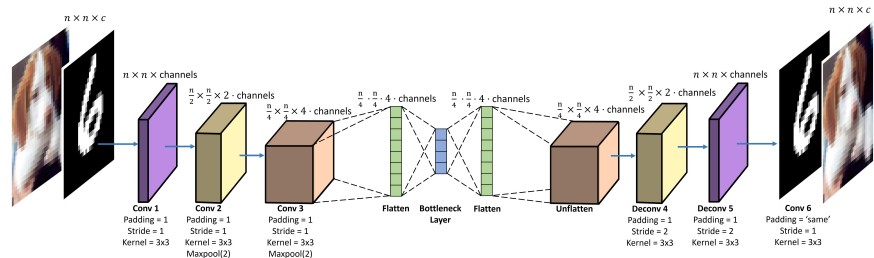

Figure 17: Demonstration of CNN AE model structure. $n, c$ represent the width, height and number of channels of each dataset. For MNIST, $n = 28, c = 1$ and for CIFAR-10, $n = 32, c = 3$. *Channels* represent the number of channels in each layer of the CNN AE.

**Loss function.** All AEs are trained with the mean squared error (MSE) loss function:

$$\text{MSE} = \frac{1}{N} \sum_{i=1}^{N} \|\boldsymbol{y}_i - \hat{\boldsymbol{y}}_i\|_2^2.$$

Where $N$ is the number of training data samples, $\boldsymbol{y}_i$ is the true vector, and $\hat{\boldsymbol{y}}_i$ is the predicted vector. Due to contamination in the training dataset, the norm of train samples tends to be higher than that of the clean test samples. As the MSE loss is not scale-invariant, we opt to normalize both train and test losses only after the training process is complete, using $\frac{1}{N} \sum_{i=1}^{N} \|\boldsymbol{y}_i - \bar{\boldsymbol{y}}\|_2^2$, Where $\bar{\boldsymbol{y}}$ is the mean of $\{\boldsymbol{y}_i\}_{i=1}^{N}$. This strategy enables us to continue utilizing the MSE loss function while facilitating a fair and meaningful comparison between train and test losses.

**Results.** Ensuring the robustness of the findings across various model initializations and enhancing their reliability, all figures combine several results of different random seeds. The bolded curves in each figure represent the average across these results, and the transparent curve around each bolded curve represents the $\pm 1$ standard error from the mean.

**Environments and Computational Time.** All experiments were conducted on NVIDIA RTX 6000 Ada Generation with 47988 MiB, NVIDIA GeForce RTX 3080 with 10000 MiB, Tesla V100-SXM2-32GB with 34400 MiB, and NVIDIA GeForce GTX 1080 Ti with 11000 MiB.

Each result in Figure 3 represents an average over 10 seeds. The hidden layer sizes for the subspace data model range from 4 to 500 with a step size of 4, and for the single-cell RNA data, they range from 10 to 500 with a step size of 10, and from 500 to 3000 with a step size of 50. This results in 125 and 110 models trained for each dataset, respectively. Figure 3(a) illustrates 10 different sample noise levels, requiring the training of $125 \times 10 \times 10 = 12{,}500$ models. Similarly, Figure 3(b) depicts 4 different sample noise levels, corresponding to $110 \times 10 \times 4 = 4{,}400$ trained models. In total, 16,900 models, each with up to 8 million parameters trained on 5,000 data points were needed to obtain the results. In Appendix F.3, Figure 35, we present results for CNNs including between 1 to 64 channels trained on 5,000 images from the MNIST dataset including 5 levels of sample noise and 6 levels of feature noise for 10 different seeds. These experiments require training $64 \times 11 \times 10 = 7{,}040$ models with up to 13 million parameters. Each evaluation of a specific experiment takes several days if trained on the NVIDIA RTX 6000 and weeks if trained on the other mentioned GPUs to obtain the results.

## B SNR Calculations

In this section, we will outline our approach for calculating the signal-to-noise ratio (SNR) for all experiments involving the addition of noise. Initially, we convert the SNR from decibels to linear SNR using the formula:

$$\text{SNR} = 10^{\left(\frac{\text{SNR[dB]}}{20}\right)}. \tag{2}$$

We have a closed-form equation for the subspace data model to determine the scalar $\beta$ required to multiply the train samples and achieve the desired linear SNR value. We use the fact that both train and noise are

sampled from an i.i.d. normal distribution and calculate $\beta$ for the sample noise, feature noise, domain shift, and anomalies.

**Notations**:
$\boldsymbol{z} - d \times 1$ vector. Represents a vector in a latent space of size $d$.
$\boldsymbol{H} - D \times d$ matrix. Represents a random matrix to project $\boldsymbol{z}$ from a $d$ dimensional space into a higher-dimensional space ($D > d$).
$\boldsymbol{\epsilon} - D \times 1$ vector. Represents the noise added to a vector with $D$ dimensions.

For the scenario of sample noise, where a particular sample is affected by noise across all its features:

$$\text{SNR}^2 = \frac{E[\|\beta \boldsymbol{H}\boldsymbol{z}\|_2^2]}{E[\|\boldsymbol{\epsilon}\|_2^2]} = \frac{E[\beta^2 \boldsymbol{z}^\top \boldsymbol{H}^\top \boldsymbol{H}\boldsymbol{z}]}{E[\boldsymbol{\epsilon}^\top \boldsymbol{\epsilon}]} = \frac{\beta^2 E_{\boldsymbol{z}}[E_{\boldsymbol{H}|\boldsymbol{z}}[\boldsymbol{z}^\top \boldsymbol{H}^\top \boldsymbol{H}\boldsymbol{z}|\boldsymbol{z}]]}{E\left[\sum_{i=1}^D \boldsymbol{\epsilon}_i^2\right]} \underset{(1)}{=} \tag{3}$$

$$\frac{\beta^2 E_{\boldsymbol{z}}[\boldsymbol{z}^\top E_{\boldsymbol{H}|\boldsymbol{z}}[\boldsymbol{H}^\top \boldsymbol{H}]\boldsymbol{z}]}{D} \underset{(2)}{=} \frac{\beta^2 E_{\boldsymbol{z}}[\boldsymbol{z}^\top D \cdot \mathbb{I}_{d \times d}\boldsymbol{z}]}{D} = \frac{\beta^2 \cdot D \cdot E_{\boldsymbol{z}}[\boldsymbol{z}^\top \boldsymbol{z}]}{D} = \beta^2 E\left[\sum_{i=1}^d \boldsymbol{z}_i^2\right] \underset{(1)}{=} \beta^2 \cdot d.$$

Isolating $\beta$, we get that $\beta = \frac{\text{SNR}}{\sqrt{d}}$.

(1) Given a vector $\boldsymbol{a} \sim \mathcal{N}(0, \mathbb{I}_D)$ of $D$ i.i.d. samples, $E\left[\sum_{i=1}^D \boldsymbol{a}_i^2\right] = \sum_{i=1}^D E[\boldsymbol{a}_i^2] = \sum_{i=1}^D 1 = D$.

(2) Given a matrix $\boldsymbol{M} \sim \mathcal{N}(0, \mathbb{I}_D)$ of size $D \times D$ where all entries are i.i.d., then

$$E[\boldsymbol{M}^\top \boldsymbol{M}] \quad = \quad E\begin{bmatrix} \boldsymbol{M}_{1,1}^2 + \cdots + \boldsymbol{M}_{D,1}^2 & \cdots & \boldsymbol{M}_{1,1}\boldsymbol{M}_{1,D} + \cdots + \boldsymbol{M}_{D,1}\boldsymbol{M}_{D,D} \\ \vdots & \ddots & \vdots \\ \boldsymbol{M}_{1,1}\boldsymbol{M}_{1,D} + \cdots + \boldsymbol{M}_{D,1}\boldsymbol{M}_{D,D} & \cdots & \boldsymbol{M}_{1,D}^2 + \cdots + \boldsymbol{M}_{D,D}^2 \end{bmatrix} \quad =$$

$$\begin{bmatrix} D & \cdots & 0 \\ \vdots & \ddots & \vdots \\ 0 & \cdots & D \end{bmatrix} = D \cdot \mathbb{I}_{D \times D}.$$

For the scenario of feature noise, each train sample has only $D \cdot p$ noisy features, meaning the noise vector contains values for only $D \cdot p$ entries. Consequently, $\beta$ is determined by $\sqrt{\frac{p}{d}} \cdot \text{SNR}$. For practitioners who want to explore the scenario involving domain shift, where the source and target are noisy, note that the matrix responsible for projecting $\boldsymbol{z}_{test}$ into a higher-dimensional space is denoted as $\boldsymbol{H}'' = \boldsymbol{H} + s \cdot \boldsymbol{H}'$ where $\boldsymbol{H}'$ is sampled from a standard normal distribution $\mathcal{N}(0, \mathbb{I})$ and both $\boldsymbol{H}$ and $\boldsymbol{H}'$ are i.i.d. Consequently, $\boldsymbol{H}''_{ij} \sim \mathcal{N}(0, 1 + s^2)$. Substituting $\boldsymbol{H}$ with $\boldsymbol{H}''$ in equation equation 3, we find that $E_{\boldsymbol{H}''|\boldsymbol{z}}[\boldsymbol{H}''^\top \boldsymbol{H}''] = D \cdot (1 + s^2) \cdot \mathbb{I}_d$, leading to $\text{SNR}^2 = (1 + s^2) \cdot \beta^2 \cdot d$, therefore $\beta = \frac{\text{SNR}}{\sqrt{(s^2+1)d}}$. In other words, since the covariance matrix of $\boldsymbol{H}''$ is $(1 + s^2)\mathbb{I}$, we need to make sure we first normalize the matrix by $\sqrt{1 + s^2}$ to maintain the identity covariance matrix.

For other datasets, such as the single-cell RNA dataset, we normalize each sample $\boldsymbol{x}$ by its norm $\|\boldsymbol{x}\|$, and similarly normalize each noise vector $\boldsymbol{\epsilon}$, yielding: $\hat{\boldsymbol{x}} = \frac{\boldsymbol{x}}{\|\boldsymbol{x}\|}$ and $\hat{\boldsymbol{\epsilon}} = \frac{\boldsymbol{\epsilon}}{\|\boldsymbol{\epsilon}\|}$. This ensures that the ratio $\frac{\hat{\boldsymbol{x}}}{\hat{\boldsymbol{\epsilon}}}$ equals 1. By employing equation equation 2, we attain the intended linear SNR factor $\beta$, and then scale down $\hat{\boldsymbol{\epsilon}}$ by $\beta$, yielding $\hat{\boldsymbol{\epsilon}}_{scaled} = \frac{\hat{\boldsymbol{\epsilon}}}{\beta}$. This guarantees that the linear SNR is $\frac{\hat{\boldsymbol{x}}}{\hat{\boldsymbol{\epsilon}}_{scaled}} = \beta$.

## C Proof of Theorem 5.2

**Assumption C.1.** Let $\boldsymbol{X}^{D \times N} = (\boldsymbol{x}_1, \boldsymbol{x}_2, \ldots, \boldsymbol{x}_N)$ represent the training set, with each column $\boldsymbol{x}_i$ corresponding to a data point according to the formulation in equation (1). Assume that the number of samples $N$ is large enough such that the rank of the data matrix $\boldsymbol{X}$ is almost surely equal to the number of features $D$. This assumption is reasonable due to the noise added to many samples.

**Theorem 4.1** [No double descent in linear AEs]  *Let $\Phi_a(\cdot;\boldsymbol{\theta})$, $\Phi_b(\cdot;\boldsymbol{\theta}) : \mathbb{R}^D \to \mathbb{R}^D$ be two linear AEs, each with $L \geq 2$ layers and architectures defined by $\mathcal{D}_a$ and $\mathcal{D}_b$ (refer to section 5 for definition), with bottleneck sizes $m_a$ and $m_b$, where $m_a < m_b \leq D$. Let $\boldsymbol{\theta}_a^{opt}$ and $\boldsymbol{\theta}_b^{opt}$ be the optimal parameters under assumption C.1. Let d denote the dimension of the support of $\mathcal{P}_{\tilde{\boldsymbol{x}}}$ (the dimension of the span of the samples $\tilde{\boldsymbol{x}} \sim \mathcal{P}_{\tilde{\boldsymbol{x}}}$). Then, the following holds,*

$$\mathcal{G}\left(\mathcal{D}_b; \boldsymbol{\theta}_b^{opt}, \mathcal{P}_{\tilde{\boldsymbol{x}}}\right) \leq \mathcal{G}\left(\mathcal{D}_a; \boldsymbol{\theta}_a^{opt}, \mathcal{P}_{\tilde{\boldsymbol{x}}}\right),$$

*with the inequality replaced with equality if: (1) $m_a = m_b$, even if $\mathcal{D}_a \neq \mathcal{D}_b$, or (2) $m_a, m_b \geq D$.*

*Proof.* Recall that the models are trained using noisy data sampled from $\mathcal{P}_{\boldsymbol{x}}$ and tested on clean data drawn from $\mathcal{P}_{\tilde{\boldsymbol{x}}}$ to see if there is noise memorization (high test loss) or signal learning (low test loss). For simplicity of notation, we will omit the attributions $a$ and $b$ unless explicitly needed. Without the loss of generality, we denote the training set by $\boldsymbol{X}^{D \times N} = (\boldsymbol{x}_1, \boldsymbol{x}_2, \ldots, \boldsymbol{x}_N)$ where the matrix column $\boldsymbol{x}_i^{D \times 1}$ are w.r.t formulation (1). Recall that $\text{rank}(\boldsymbol{X}) = D$, with probability 1 under assumption C.1. Let $\boldsymbol{\Sigma} \in \mathbb{R}^{D \times D}$ represent the covariance matrix, defined as $\boldsymbol{\Sigma} = \boldsymbol{X}\boldsymbol{X}^\top$. Let its singular value decomposition (SVD) be expressed as $\boldsymbol{\Sigma} = \boldsymbol{U}\boldsymbol{\Lambda}\boldsymbol{U}^\top$, and similarly $\boldsymbol{\Sigma}_{\tilde{\boldsymbol{x}}}^{D \times D} = \tilde{\boldsymbol{U}}\boldsymbol{\Omega}\tilde{\boldsymbol{U}}^\top$. Note that the eigenvectors of $\boldsymbol{\Sigma}$, which are the columns of $\boldsymbol{U}$, are arranged in descending order corresponding to their associated eigenvalues in $\boldsymbol{\Lambda}$. By Lemma C.2, since $\text{rank}(\boldsymbol{X}) = D \geq m$, then for the optimal solution, $\boldsymbol{\theta}^{opt} = \text{argmin}_{\boldsymbol{\theta}}\mathcal{L}(\boldsymbol{\theta}; \boldsymbol{X})$, we have that $\text{rank}\left(\boldsymbol{W}_{L-1}^{opt} \cdots \boldsymbol{W}_2^{opt}\right) = m$, means that the rank of the inner layers multiplication is exactly the size of the bottleneck. By Theorem 3.2, proof Case II in (Kawaguchi, 2016), all local minima of the considered optimization problem are global minima, and the expression of a global minimum of the training optimization, namely

$$\boldsymbol{\theta}^{opt} = \text{argmin}_{\boldsymbol{\theta}}\mathcal{L}(\boldsymbol{\theta}; \boldsymbol{X}) = \text{argmin}_{\boldsymbol{\theta}}\frac{1}{N}\|\Phi(\boldsymbol{X}; \boldsymbol{\theta}) - \boldsymbol{X}\|_F^2,$$

is given by

$$\boldsymbol{\theta}^{opt} = \boldsymbol{U}_m\boldsymbol{U}_m^\top\boldsymbol{X}\boldsymbol{X}^\top\left(\boldsymbol{X}\boldsymbol{X}^\top\right)^-,$$

where $\boldsymbol{U}_m^{D \times m}$, is the matrix containing the first $m$ eigenvectors of the full eigenbasis $\boldsymbol{U}^{D \times D}$ of $\boldsymbol{X}\boldsymbol{X}^\top$, namely $\boldsymbol{U}_m = [\boldsymbol{u}_1, \ldots, \boldsymbol{u}_m]$ (those corresponding to the $m$ largest eigenvalues), and the operator $(\cdot)^-$ stands for the pesudo inverse. Since $\boldsymbol{\Sigma}^{D \times D} = \boldsymbol{X}\boldsymbol{X}^\top$ is invertible ($\text{rank}(\boldsymbol{X}) = D$), we have

$$\boldsymbol{\theta}^{opt} = \boldsymbol{U}_m\boldsymbol{U}_m^\top,$$

and consequently,

$$\Phi(\boldsymbol{X}; \boldsymbol{\theta}) = \boldsymbol{U}_m\boldsymbol{U}_m^\top\boldsymbol{X}.$$

Since both, $\boldsymbol{U}^{D \times D}$ and $\tilde{\boldsymbol{U}}^{D \times D}$ are orthonormal and span $\mathbb{R}^D$ (by the SVD property), there exists an orthogonal matrix $\boldsymbol{R}^{D \times D}$ such that $\boldsymbol{U} = \boldsymbol{R}\tilde{\boldsymbol{U}}$. This matrix $\boldsymbol{R} = \boldsymbol{U}\tilde{\boldsymbol{U}}^\top$ is unitary by construction, satisfying $\boldsymbol{R}^\top\boldsymbol{R} = \mathbb{I}_{D \times D}$, and represents a rotation or reflection aligning the test and training eigenspaces. Thus $\boldsymbol{U}$ and $\tilde{\boldsymbol{U}}$ are related by this $\boldsymbol{R}$, such that

$$\boldsymbol{U} = \boldsymbol{R}\tilde{\boldsymbol{U}}.$$

Although the truncated matrices $\boldsymbol{U}_m$ and $\tilde{\boldsymbol{U}}_m$ generally span different $m$-dimensional subspaces, both reside within the same ambient $D$-dimensional space, thus enabling such an orthonormal transformation $\boldsymbol{R}$.

Calculating the generalization error,

$$\begin{aligned}
\mathcal{G}(\mathcal{D}; \boldsymbol{\theta}^{opt}, \mathcal{P}_{\tilde{\boldsymbol{x}}}) &= \mathbb{E}\left[\|\Phi(\tilde{\boldsymbol{x}}; \boldsymbol{\theta}^{opt}) - \tilde{\boldsymbol{x}}\|_2^2\right] \\
&= \mathbb{E}\left[\|\boldsymbol{U}_m\boldsymbol{U}_m^\top\tilde{\boldsymbol{x}} - \tilde{\boldsymbol{x}}\|_2^2\right] \\
&= \mathbb{E}\left[\|(\mathbb{I} - \boldsymbol{U}_m\boldsymbol{U}_m^\top)\tilde{\boldsymbol{x}}\|_2^2\right] \\
&= \mathbb{E}\left[\tilde{\boldsymbol{x}}^T(\mathbb{I} - \boldsymbol{U}_m\boldsymbol{U}_m^T)^T(\mathbb{I} - \boldsymbol{U}_m\boldsymbol{U}_m^T)\tilde{\boldsymbol{x}}\right] \\
&= \mathbb{E}\left[\text{Tr}\left((\mathbb{I} - \boldsymbol{U}_m\boldsymbol{U}_m^T)\tilde{\boldsymbol{x}}\tilde{\boldsymbol{x}}^\top\right)\right] \\
&= \text{Tr}\left[(\mathbb{I} - \boldsymbol{U}_m\boldsymbol{U}_m^T)\mathbb{E}\left[\tilde{\boldsymbol{x}}\tilde{\boldsymbol{x}}^\top\right]\right] \\
&= \text{Tr}\left[(\mathbb{I} - \boldsymbol{U}_m\boldsymbol{U}_m^T)\boldsymbol{\Sigma}_{\tilde{\boldsymbol{x}}}\right]
\end{aligned}$$

$$
\begin{aligned}
&= \operatorname{Tr}\left[\boldsymbol{\Sigma}_{\tilde{\boldsymbol{x}}}\right] - \operatorname{Tr}\left[\boldsymbol{U}_m \boldsymbol{U}_m^T \boldsymbol{\Sigma}_{\tilde{\boldsymbol{x}}}\right] \\
&= \operatorname{Tr}\left[\boldsymbol{\Sigma}_{\tilde{\boldsymbol{x}}}\right] - \operatorname{Tr}\left[\boldsymbol{R}\tilde{\boldsymbol{U}}_m \tilde{\boldsymbol{U}}_m^\top \boldsymbol{R}^\top \boldsymbol{\Sigma}_{\tilde{\boldsymbol{x}}}\right] \\
&= \operatorname{Tr}(\boldsymbol{\Sigma}_{\tilde{\boldsymbol{x}}}) - \mathbb{E}\left[\left\|\tilde{\boldsymbol{U}}_m^\top \boldsymbol{R}^\top \tilde{\boldsymbol{x}}\right\|_2^2\right],
\end{aligned}
\tag{4}
$$

where $\boldsymbol{\Sigma}_{\tilde{\boldsymbol{x}}}^{D \times D} := \mathbb{E}\left[\tilde{\boldsymbol{x}}\tilde{\boldsymbol{x}}^\top\right]$. Now, since there exists at least $d$ different indexes $j \in [D]$ for which $\boldsymbol{u}_j^\top \boldsymbol{R} \neq 0$ (otherwise the latent dimension of the test-set $\tilde{\boldsymbol{x}}$ would be strictly lower than $d$ in contradiction), then for $m_a + 1 < d$ and $m_b + 1 < d$, $\mathbb{E}\left[\left\|\tilde{\boldsymbol{U}}_{m_a}^\top \boldsymbol{R}^\top \tilde{\boldsymbol{x}}\right\|_2^2\right]$, $\mathbb{E}\left[\left\|\tilde{\boldsymbol{U}}_{m_b}^\top \boldsymbol{R}^\top \tilde{\boldsymbol{x}}\right\|_2^2\right] > 0$. Hance, since the larger is $m$ in (4) the lower is $\mathcal{G}$, we have

$$
\mathcal{G}\left(\mathcal{D}_b; \boldsymbol{\theta}_{\boldsymbol{b}}^{\mathrm{opt}}, \mathcal{P}_{\tilde{\boldsymbol{x}}}\right) \leq \mathcal{G}\left(\mathcal{D}_a; \boldsymbol{\theta}_{\boldsymbol{a}}^{\mathrm{opt}}, \mathcal{P}_{\tilde{\boldsymbol{x}}}\right),
$$

and obviously, if $m_a, m_b \geq D$, then

$$
\mathcal{G}\left(\mathcal{D}_b; \boldsymbol{\theta}_{\boldsymbol{b}}^{\mathrm{opt}}, \mathcal{P}_{\tilde{\boldsymbol{x}}}\right) = \mathcal{G}\left(\mathcal{D}_a; \boldsymbol{\theta}_{\boldsymbol{a}}^{\mathrm{opt}}, \mathcal{P}_{\tilde{\boldsymbol{x}}}\right) = 0.
$$

$\square$

**Lemma C.2.** *Consider the pre-trained linear AE, $\Phi(\cdot; \boldsymbol{\theta}^{opt}) : \mathbb{R}^D \to \mathbb{R}^D$, of $L \geq 2$ layers, where $\boldsymbol{\theta}^{opt} \equiv \left\{\boldsymbol{W}_1^{d_1 \times D}, \ldots, \boldsymbol{W}_{L-1}^{d_{L-1} \times d_{L-2}}, \boldsymbol{W}_L^{D \times d_{L-1}}\right\}$, for which its bottleneck size holds $m(\leq D)$, and a training set $\boldsymbol{X}$, of any dimension satisfies $rank(\boldsymbol{X}) \geq m$, then $\operatorname{rank}(\boldsymbol{W}_{L-1} \cdots \boldsymbol{W}_2) = m$.*

*Proof.* Let $\boldsymbol{X} \in \mathbb{R}^{D \times N}$ be the training data, with $\operatorname{rank}(\boldsymbol{X}) \geq m$. The reconstruction is given by

$$
\hat{\boldsymbol{X}} = \boldsymbol{W}_L \boldsymbol{W}_{L-1} \cdots \boldsymbol{W}_2 \boldsymbol{W}_1 \boldsymbol{X}.
$$

Accordingly, the reconstruction loss is

$$
\mathcal{L} = \|\boldsymbol{X} - \hat{\boldsymbol{X}}\|_F^2 = \|\boldsymbol{X} - \boldsymbol{W}_L \boldsymbol{W}_{L-1} \cdots \boldsymbol{W}_2 \boldsymbol{W}_1 \boldsymbol{X}\|_F^2.
$$

If $\operatorname{rank}(\boldsymbol{W}_{L-1} \cdots \boldsymbol{W}_2) = d < m$, then the rank of $\hat{\boldsymbol{X}}$ is at most $d$. This means that $\boldsymbol{W}_{L-1} \cdots \boldsymbol{W}_2 \boldsymbol{W}_1$ maps $\boldsymbol{X}$ into a subspace of dimension $d < m$. However, the optimal low-rank approximation of $\boldsymbol{X}$ that minimizes the reconstruction loss is achieved by projecting $\boldsymbol{X}$ onto the subspace spanned by its top $m$ singular vectors (the rank-$m$ approximation). By the Eckart-Young-Mirsky theorem (Eckart & Young, 1936),

$$
\mathcal{L}_{\mathrm{optimal}} = \|\boldsymbol{X} - \boldsymbol{X}_m\|_F^2,
$$

where $\boldsymbol{X}_m$ is the rank-$m$ approximation of $\boldsymbol{X}$. If $\operatorname{rank}(\boldsymbol{W}_{L-1} \cdots \boldsymbol{W}_2) < m$, the projection would not span the top $m$ singular vectors, leading to

$$
\mathcal{L} > \mathcal{L}_{\mathrm{optimal}},
$$

contradicting the assumption that weights are optimal. $\square$

*Remark* C.3 (Generalization for noisy test data). Throughout the paper, we trained non-linear models on contaminated datasets, drawn from $\mathcal{P}_{\boldsymbol{x}}$ and tested on clean data sampled from $\mathcal{P}_{\tilde{\boldsymbol{x}}}$ to isolate the effect on the learned model, showing noise memorization (high test loss) versus signal learning (low test loss) and observed model, epoch, and sample-wise multiple descents. We followed a similar approach as in supervised learning, where models are trained with noisy labels and tested on clean ones (in our unsupervised scenario, we trained on noisy data and tested on clean data, similar to (Lupidi et al., 2023)). We also showed that for the case of linear models, the test loss decreases as model size increases. If we define the generalization loss **differently** than the definition in the paper, to account for noise in the test data, then the test loss does not have to

monotonically decrease when $m$ increases.

$$\begin{aligned}
\mathcal{G}'(D; \boldsymbol{\theta}^{opt}, \mathcal{P}_{\tilde{\boldsymbol{x}}}) &= \mathbb{E}\left[\left\|\Phi(\tilde{\boldsymbol{x}} + \boldsymbol{n}; \boldsymbol{\theta}^{opt}) - \tilde{\boldsymbol{x}}\right\|_2^2\right] \\
&= \mathbb{E}\left[\left\|\mathbf{U}_m\mathbf{U}_m^\top(\tilde{\boldsymbol{x}} + \boldsymbol{n}) - \tilde{\boldsymbol{x}}\right\|_2^2\right] \\
&= \mathbb{E}\left[\left\|(\mathbf{U}_m\mathbf{U}_m^\top - \mathbb{I})\tilde{\boldsymbol{x}} + \mathbf{U}_m\mathbf{U}_m^\top\boldsymbol{n}\right\|_2^2\right] \\
&= \mathbb{E}\left[\left\|(\mathbb{I} - \mathbf{U}_m\mathbf{U}_m^\top)\tilde{\boldsymbol{x}}\right\|_2^2 + \left\|\mathbf{U}_m\mathbf{U}_m^\top\boldsymbol{n}\right\|_2^2 + 2\langle(\mathbb{I} - \mathbf{U}_m\mathbf{U}_m^\top)\tilde{\boldsymbol{x}}, \mathbf{U}_m\mathbf{U}_m^\top\boldsymbol{n}\rangle\right] \\
&= \mathbb{E}\left[\left\|(\mathbb{I} - \mathbf{U}_m\mathbf{U}_m^\top)\tilde{\boldsymbol{x}}\right\|_2^2\right] + \mathbb{E}\left[\left\|\mathbf{U}_m\mathbf{U}_m^\top\boldsymbol{n}\right\|_2^2\right] \\
&= \text{Tr}\left[(\mathbb{I} - \mathbf{U}_m\mathbf{U}_m^\top)\Sigma_{\tilde{\boldsymbol{x}}}\right] + \text{Tr}\left[\mathbf{U}_m\mathbf{U}_m^\top\Sigma_{\boldsymbol{n}}\right] \\
&= \underbrace{\text{Tr}\left[(\mathbb{I} - \mathbf{U}_m\mathbf{U}_m^\top)\Sigma_{\tilde{\boldsymbol{x}}}\right]}_{\text{clean error}} + \underbrace{\text{Tr}\left[\mathbf{U}_m\mathbf{U}_m^\top\Sigma_{\boldsymbol{n}}\right]}_{\text{projected noise contribution}} \quad .
\end{aligned}$$

We used the assumption that the expectation of the noise is zero. When $m$ increases, the first term decreases while the second term increases, implying that increasing $m$ does not have to monotonically decrease this definition of the generalization error.

## D  Train Loss Results

In this section, we provide the train loss figures corresponding to each of the test losses mentioned in the main paper.

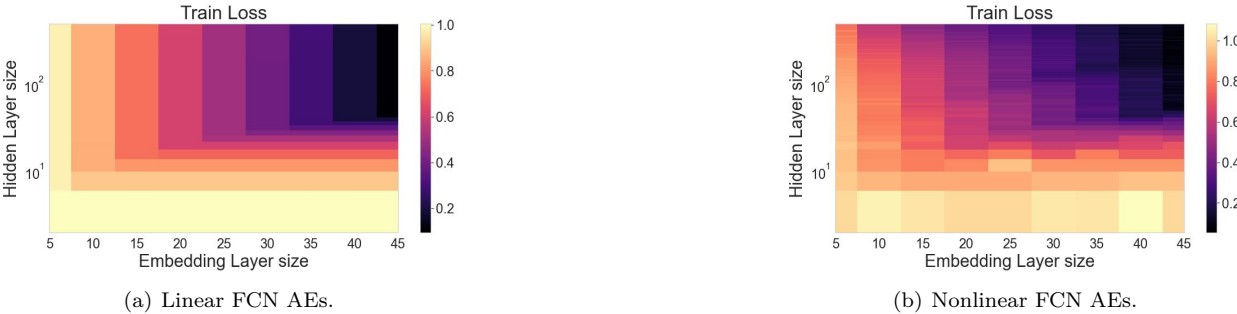

(a) Linear FCN AEs.

(b) Nonlinear FCN AEs.

Figure 18: Heatmap of train losses for both linear and nonlinear AEs.

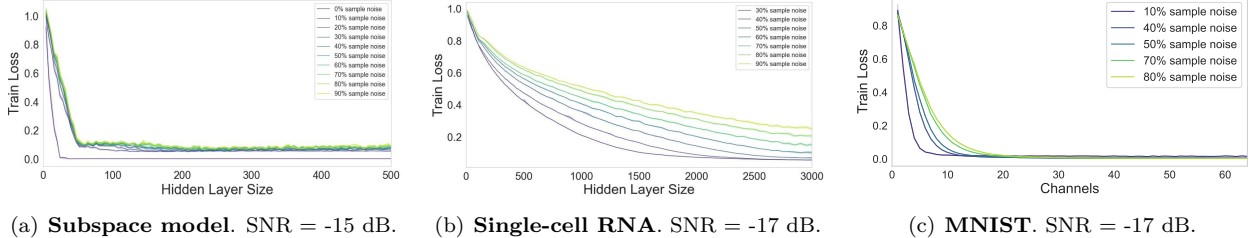

(a) **Subspace model**. SNR = -15 dB.

(b) **Single-cell RNA**. SNR = -17 dB.

(c) **MNIST**. SNR = -17 dB.

Figure 19: Model-wise train losses for the case of varying **sample noise**.

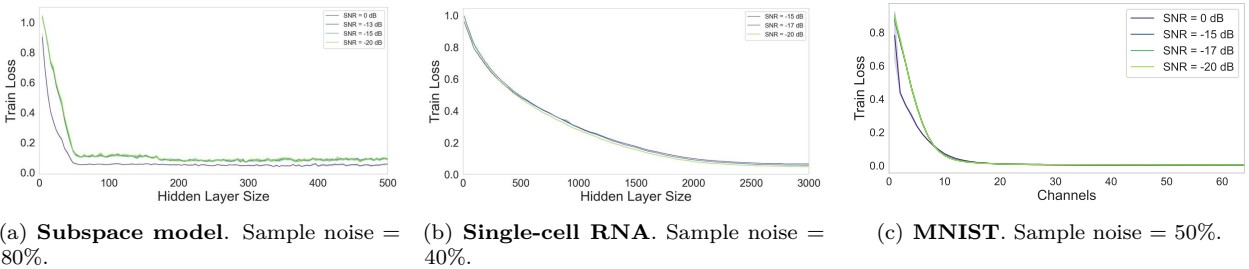

(a) **Subspace model**. Sample noise = 80%.

(b) **Single-cell RNA**. Sample noise = 40%.

(c) **MNIST**. Sample noise = 50%.

Figure 20: Model-wise train losses for the case of **noisy samples** and varying SNR.

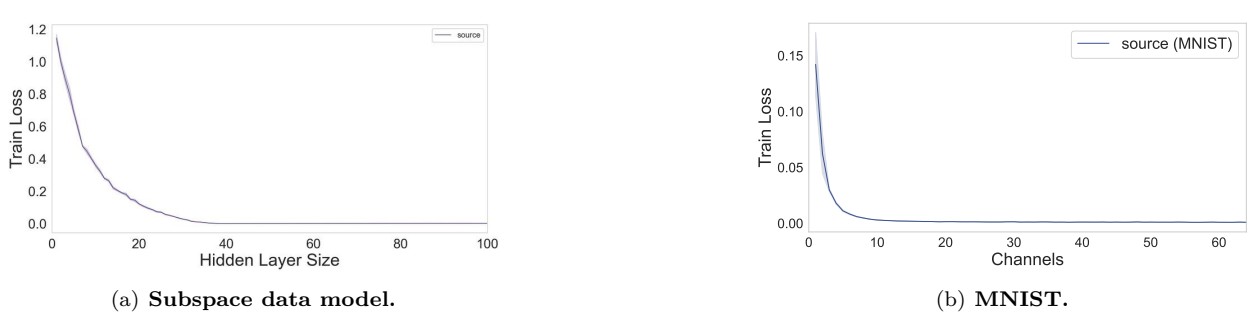

(a) **Subspace data model.**

(b) **MNIST.**

Figure 21: Train losses of source data.

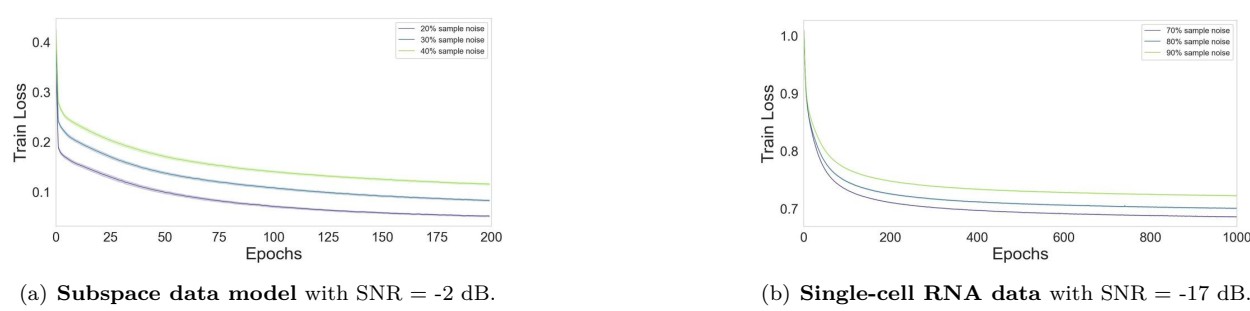

(a) **Subspace data model** with SNR = -2 dB.

(b) **Single-cell RNA data** with SNR = -17 dB.

Figure 22: Epoch-wise train losses for varying number of **noisy samples**.

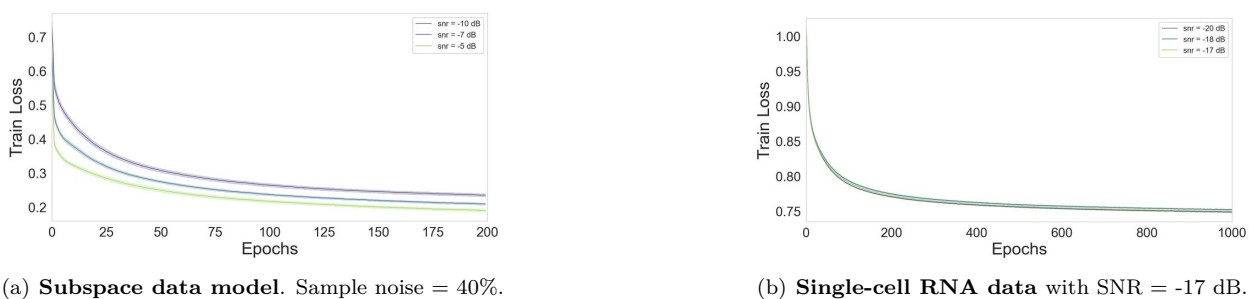

(a) **Subspace data model**. Sample noise = 40%.

(b) **Single-cell RNA data** with SNR = -17 dB.

Figure 23: Epoch-wise train losses for the case of **sample noise** influenced by the SNR.

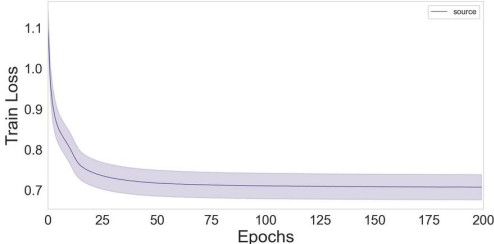
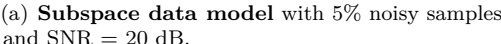
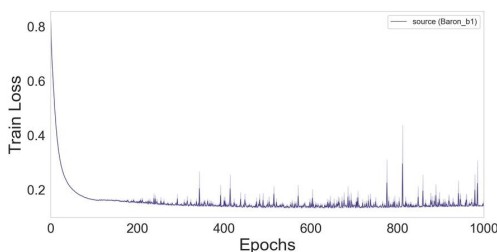

(a) **Subspace data model** with 5% noisy samples and SNR = 20 dB.

(b) **Single-cell RNA data** trained on the 'Baron' batch.

Figure 24: Epoch-wise train loss influenced by the amount of **domain shift**.

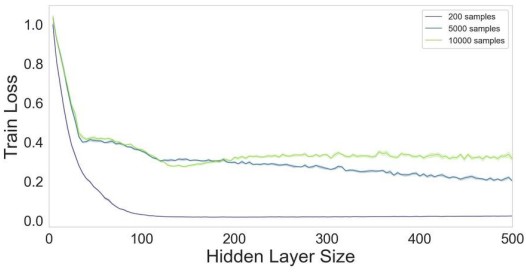

Figure 25: Train loss of the **subspace data model** with sample noise = 70% and SNR = -15 dB for different number of training samples.

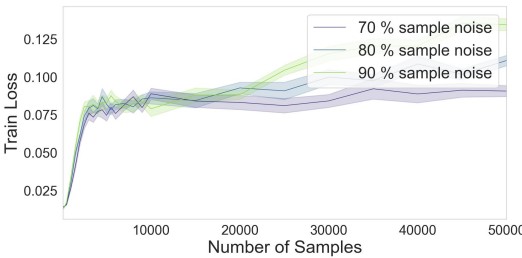

Figure 26: Train loss rises as the number of samples increases for the scenario of sample noise. Similar behavior exists for feature noise and domain shift scenarios. Train loss rises because a larger training set makes the model relatively under-parameterized, reducing training effectiveness and resulting in higher train loss. However, despite the increase in training loss with more samples, the test loss decreases, suggesting improved reconstruction of the test samples.

# E   Results for Feature Noise

In this section, we present the results for the feature noise scenario. Feature noise adds complexity since each sample contains noise in some of its features. As a result, the model never encounters samples with entirely clean features, making it unable to isolate and focus on clean data. Consequently, the model experiences difficulty in learning the correct data structure. Surprisingly, increasing feature noise actually leads to a decrease in the test loss for the single-cell RNA dataset (Figure 27(b)). This can also be observed in Appendix F.3, Figure 35(b) and Appendix F.4, Figure 46(b). Moreover, the peak shifts left as the number of noisy features rises in Figure 27(a).

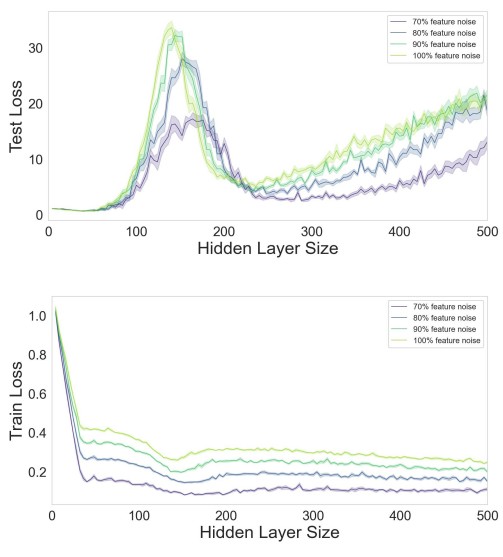
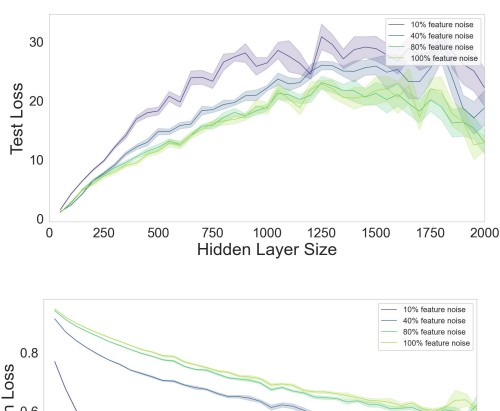
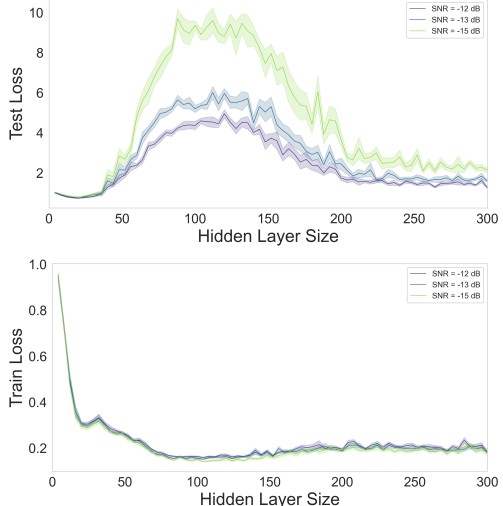
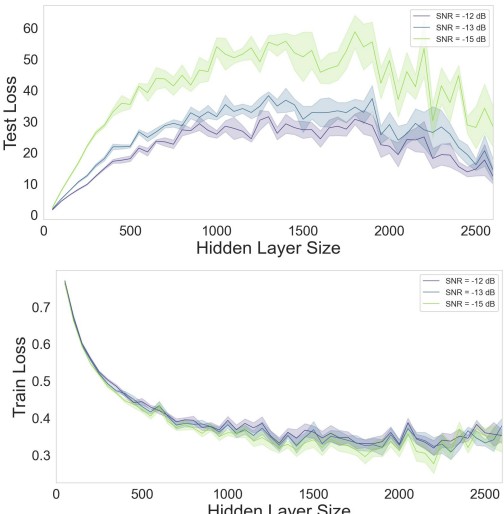

(a) Final ascent phenomenon (Xue et al., 2022) for the **Subspace data model** trained with SNR = -13 dB.

(b) Non-monotonic behavior for the **Single-cell RNA data** trained with SNR = -12 dB.

Figure 27: Test loss exhibits model-wise double descent and non-monotonic behaviors for the case of varying **feature noise**.

(a) **Subspace data model** with 40% noisy features. Beyond hidden layer of size 300, the test loss rises.

(b) **Single-cell RNA data** with 10% noisy features. Beyond a hidden layer size of 2600, the test loss continues to decrease, and the train loss increases.

Figure 28: The effect of SNR for the case of **noisy features** on the test loss curve.

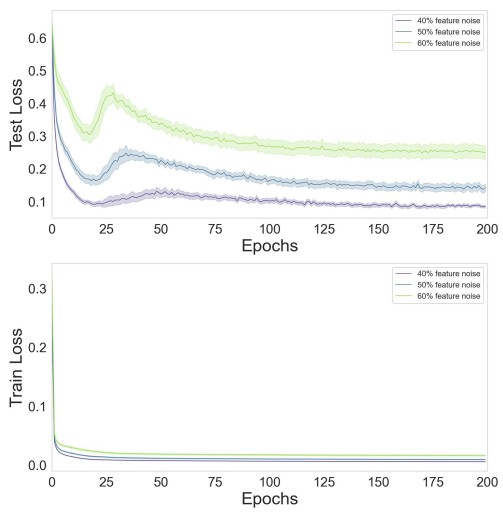

(a) **Subspace data model** with SNR = -12 dB.

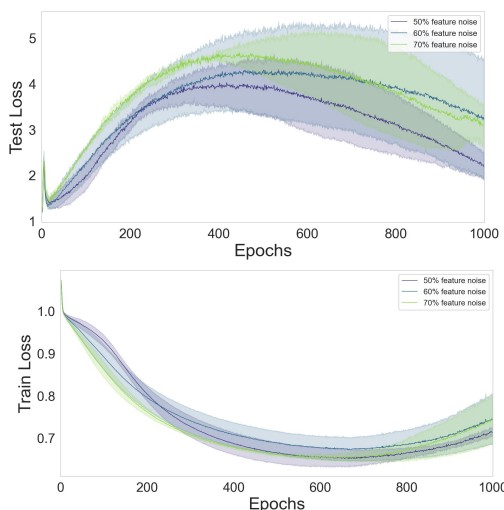

(b) **Single-cell RNA data** with SNR = -12 dB.

Figure 29: Epoch-wise double descent influenced by the number of **noisy features**.

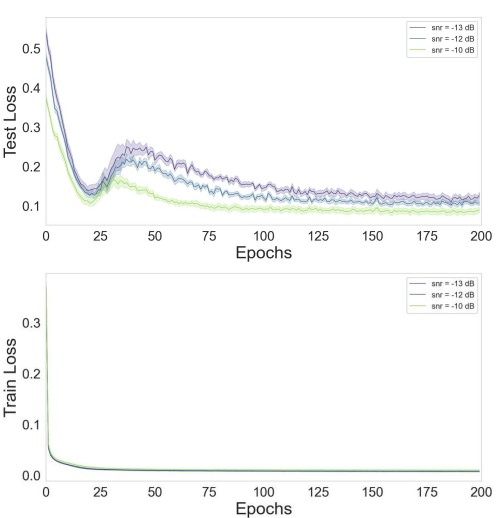

(a) **Subspace data model**. Feature noise = 80%.

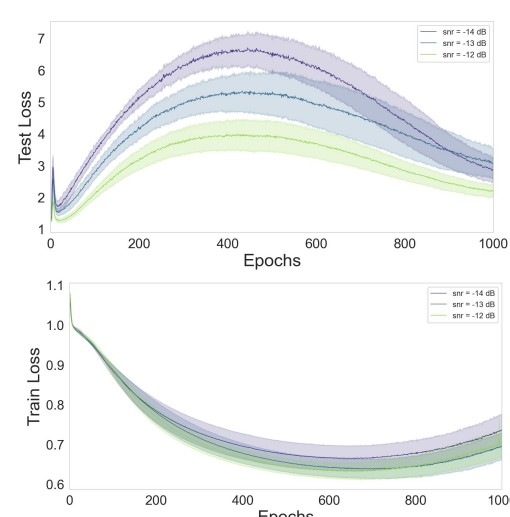

(b) **Single-cell RNA data**. Feature noise = 50%.

Figure 30: Epoch-wise double descent for the case of **feature noise** influenced by the SNR.

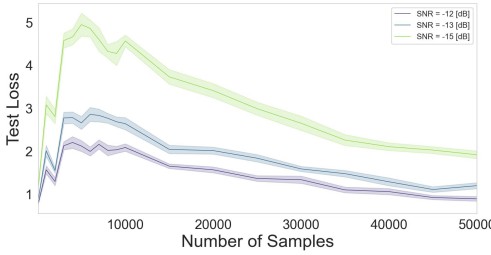

(a) Sample-wise non-monotonicity for varying **SNR**s in the scenario of **feature noise** = 90 %.

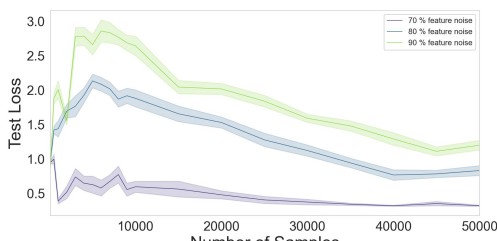

(b) Sample-wise non-monotonicity for varying number of **noisy features**. SNR = -13 dB.

Figure 31: **Sample-wise** non-monotonicity pattern for the **subspace data model** for the scenario of feature noise.

# F   Additional Experiments

## F.1   More Results for Domain Adaptation

This section presents the UMAP visualizations of the different domains for both the subspace model and single-cell RNA data in Figure 32. Results for different model sizes trained on the subspace model dataset are also reported in Figure 33.

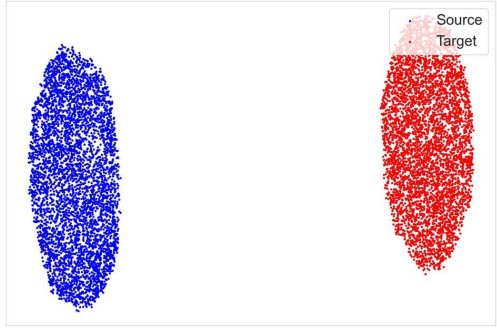
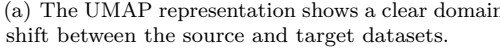

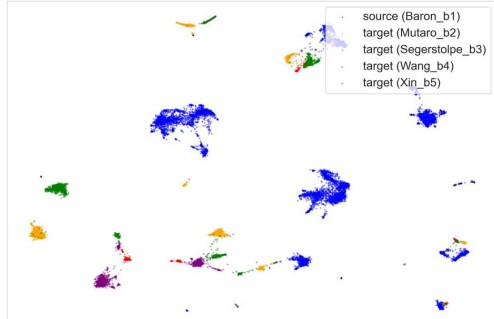

(a) The UMAP representation shows a clear domain shift between the source and target datasets.

(b) Clusters represent different cell types. Different domains are represented by different colors.

Figure 32: UMAP representations of source and target datasets for the subspace data model (a) and single-cell RNA dataset (b).

Figure 33 illustrates the results based on a similar experiment conducted in Section 7.1 for the subspace data model. As expected, the interpolating models exhibit the poorest KNN-DAT outcomes. Over-parameterized models introduce a decrease in the test loss indicating an improved reconstruction of the target data. In this scenario, we noticed that smaller models perform better than over-parameterized models based on KNN-DAT results. We think that the small size of the hidden layer (4) and the high dimensionality of the dataset (50 features) result in significant information loss in these layers. This could lead to closely clustered vectors in the embedding space, ultimately causing low KNN-DAT results. However, a hidden layer of size 4 indicates insufficient capacity to represent the signal, as shown by the high values of test and train losses in Figure 33.

## F.2   Overcomplete AEs

In this section, we show that overcomplete AEs learn the identity function, rendering them irrelevant for our study. Figure 34 is an extension of Figure 2 to overcomplete AEs, where models are trained on the subspace data model (Subsection 3.1) with $D = 50$ features. Models with embeddings and hidden layers exceeding $D$ (above the white dashed lines), making them overcomplete, achieve zero train and test reconstruction loss, confirming they learn the identity function, resulting in the absence of double descent.

## F.3   Double Descent Results For CNNs Trained on MNIST and CIFAR-10

In this section, we demonstrate that the double descent phenomenon can be reproduced in other unsupervised AE architectures. We employed the MNIST (LeCun et al., 1998) and CIFAR-10 (Krizhevsky et al., 2009) datasets and trained undercomplete CNNs as detailed in Figure 17. For the case of sample noise, the noise is added to $p \cdot 100\%$ of the images, and we present results for both MNIST and CIFAR-10, while all the other cases are presented using the MNIST dataset. For the feature noise scenario, noise is introduced to $p \cdot 100\%$ of the pixels of each image. To demonstrate the phenomenon with the presence of domain shift, the models are trained on the MNIST-M and MNIST datasets and tested on MNIST and MNIST-M, respectively. Results for model-wise double descent for varying amounts of sample and feature noise cases for the MNIST dataset are presented in Figures 3(c), 35 respectively. Varying levels of sample noise and SNRs for the CIFAR-10 dataset are presented in Figure 36.

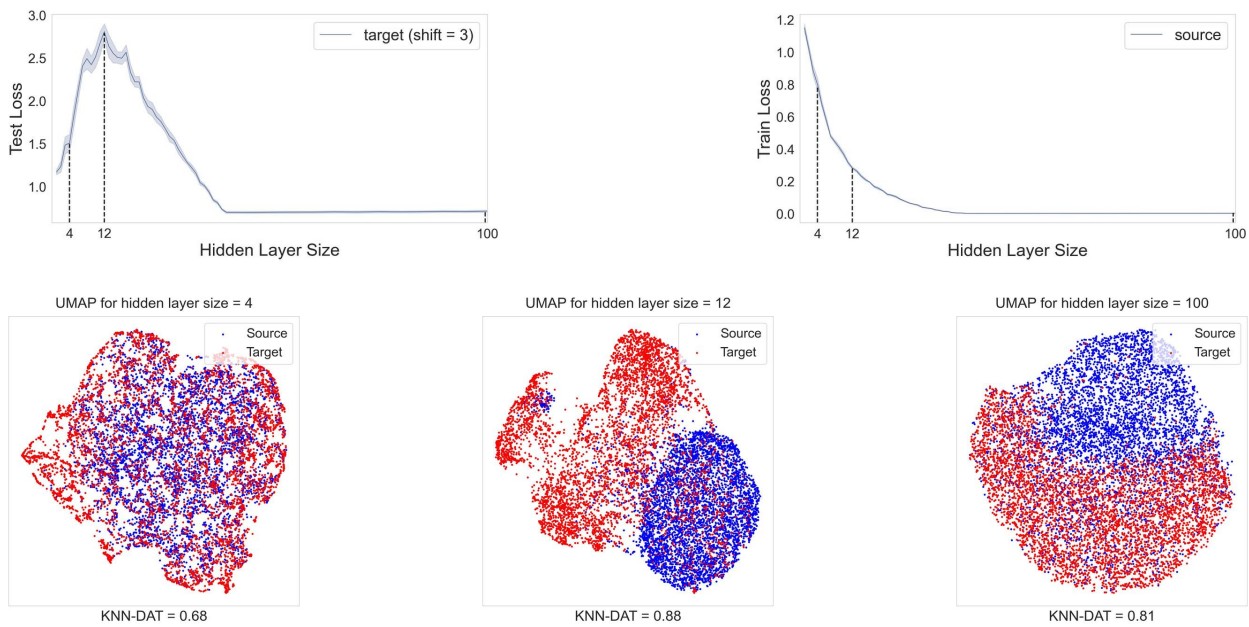

Figure 33: UMAP of the embedding vectors with a size of 45 and KNN-DAT results for different model sizes trained on the **subspace data model** for a shift of 3.

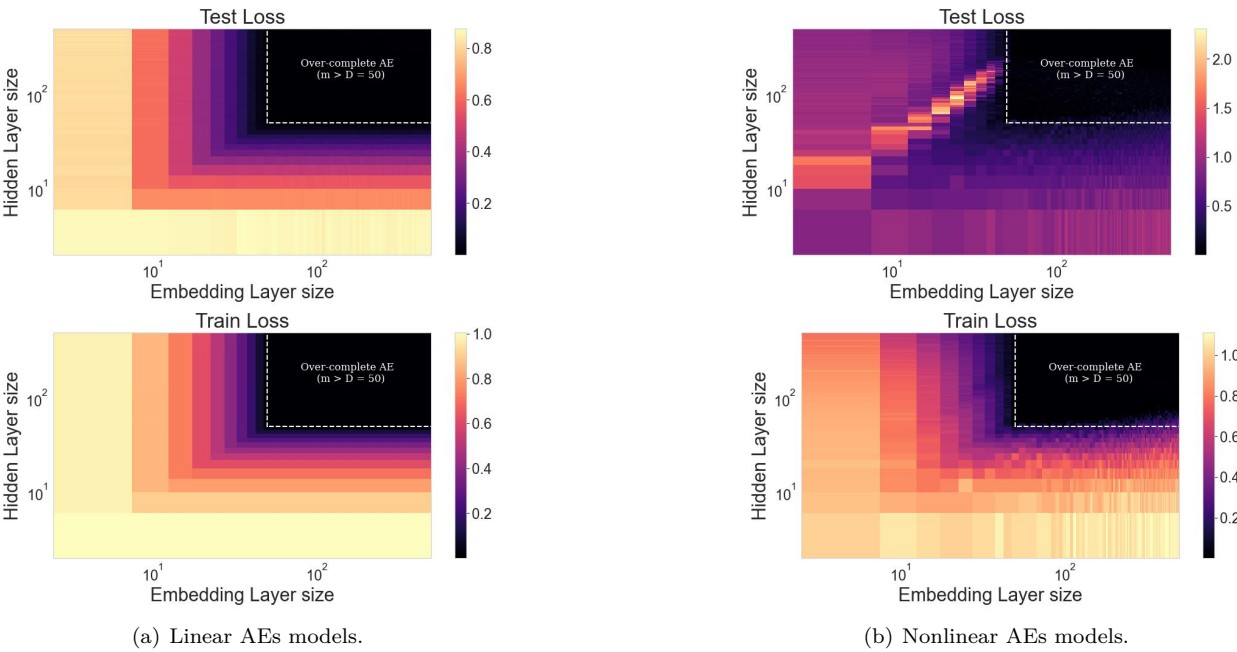

(a) Linear AEs models.

(b) Nonlinear AEs models.

Figure 34: An extension of the heatmaps presented in Figure 2 to overcomplete AEs (models above the white dashed lines) resulting in zero train and test reconstruction loss for both linear and nonlinear AEs.

In Figure 37, we show the test and train loss results (top two sub-figures) for three different models (3, 5, and 60 channels) trained on MNIST with 50% sample noise and an SNR of -15 dB and find out that over-parameterized models can reduce the noise levels in an image. The smallest model, with 3 channels, is under-parameterized. The second model, within the critical regime, with 5 channels, performs poorly, while the third is over-parameterized, containing 60 channels. Interestingly, We noticed that even though our

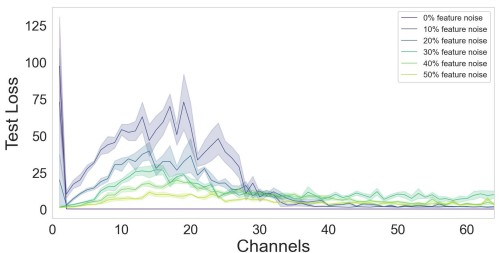 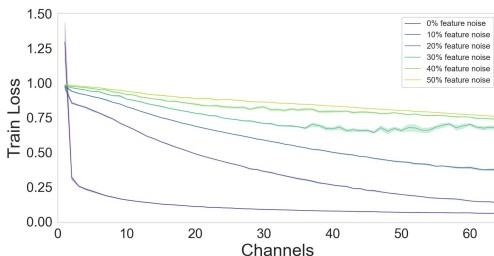

Figure 35: Model-wise double descent for CNNs trained on MNIST with varying levels of feature noise and SNR = -20 dB.

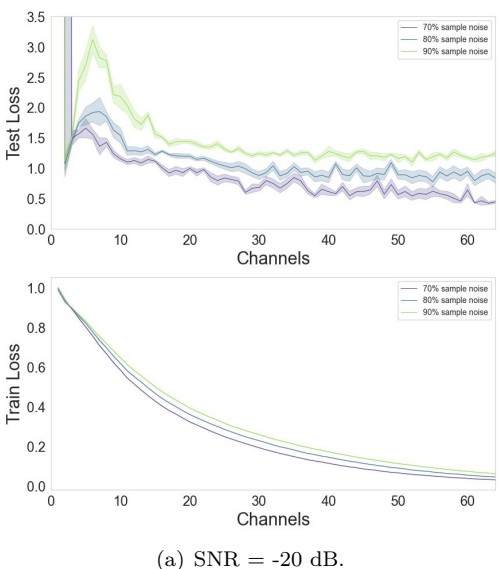 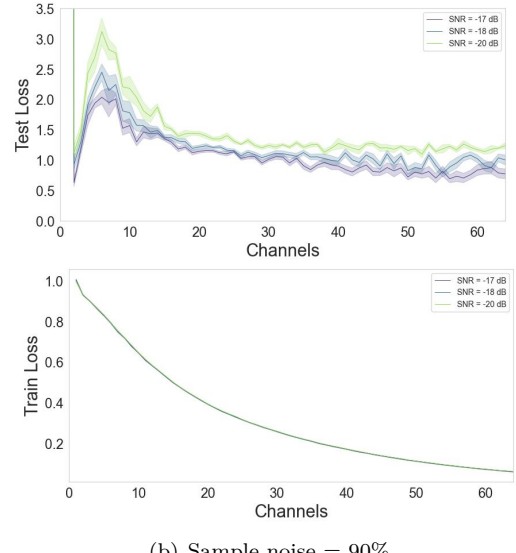

(a) SNR = -20 dB.

(b) Sample noise = 90%

Figure 36: Model-wise double descent for CNNs trained on CIFAR-10 with varying levels of sample noise (a) and SNRs (b).

AE was not trained to remove noise (as in denoising AEs (Vincent et al., 2008; 2010)), over-parameterized models were able to reduce noise to some extent. In contrast, models within the critical regime performed significantly worse.

After training, we evaluated each model by feeding it images with varying SNR values and examining the reconstructed outputs (bottom sub-figure in Figure 37). The over-parameterized model produced the best-quality reconstructed images. Following that, the under-parameterized model performed moderately well, and the model in the critical regime generated the noisiest images. This is because the critical model focused on memorizing the noise during training instead of learning the underlying signal, resulting in consistently noisy outputs. In contrast, the over-parameterized model had enough capacity to memorize the noise and learn the signal. While the under-parameterized model cleans the images better than the critical model, it still distorts some details compared to the over-parameterized model due to its limited capacity.

To quantify noise reduction, we used the Peak Signal-to-Noise Ratio (PSNR), a metric that assesses signal quality by comparing the original image to its noisy version. PSNR measures the ratio between the maximum possible value of a signal ($R^2$) and the power of the noise (MSE). Higher PSNR values indicate better quality, meaning less noise. The formula for PSNR is:

$$PSNR = 10 \cdot \log\left(\frac{R^2}{MSE(\boldsymbol{x}, f(\boldsymbol{x} + \boldsymbol{\epsilon}))}\right),$$

where $\boldsymbol{x} + \boldsymbol{\epsilon}$ represents the noisy image ($\boldsymbol{\epsilon}$ is the noise), and $\boldsymbol{x}$ is the clean version. This metric, expressed in decibels, allows us to evaluate how well each model cleans the images. As shown, the over-parameterized model

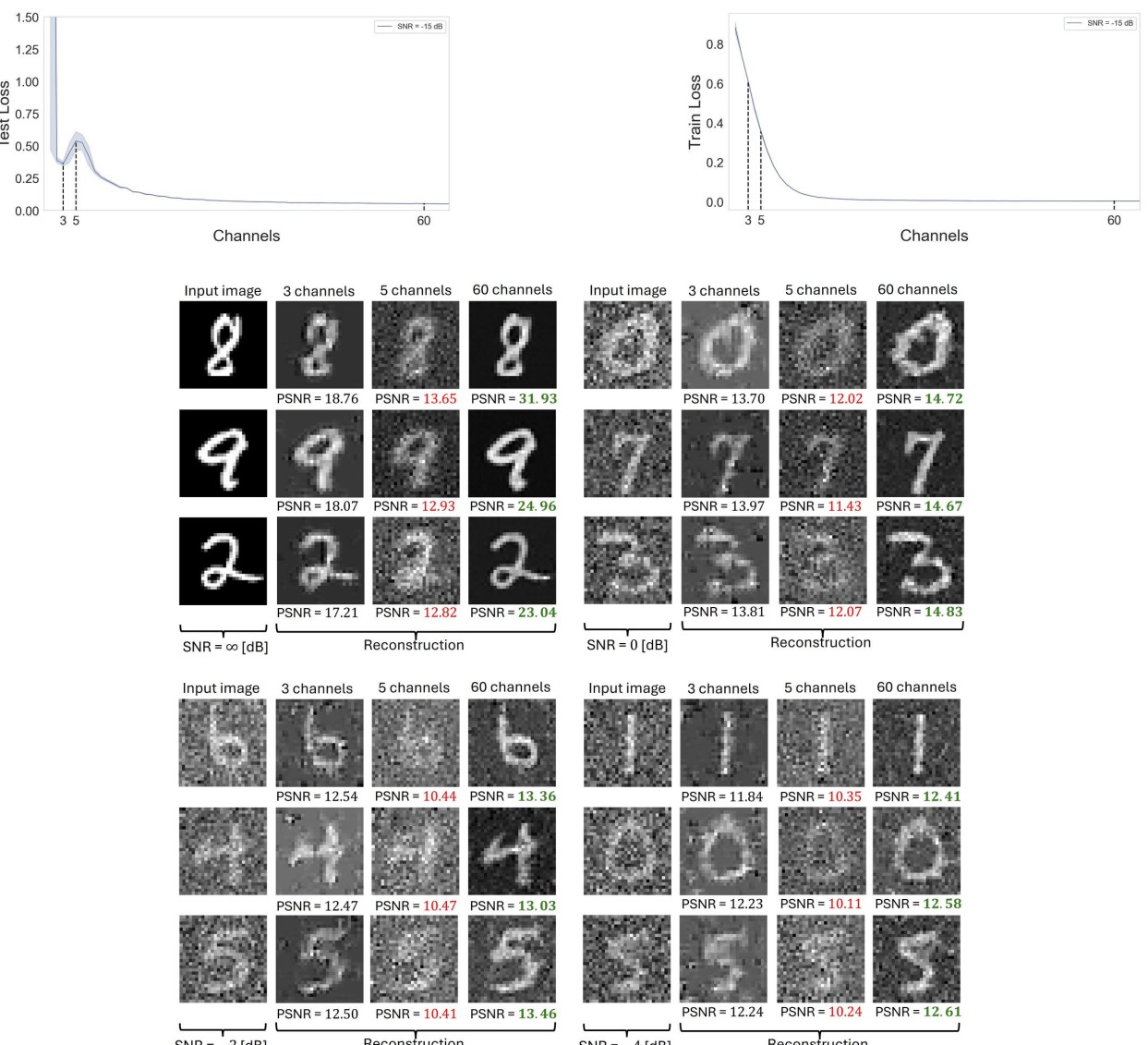

Figure 37: Models trained on 50% noisy MNIST images with SNR = -15 [dB] and tested on MNIST images with different values of SNR.

consistently achieves the highest PSNR values (highlighted in bold green), while the poorly interpolating model, which primarily memorized noise, produces the lowest PSNR values (in red). In conclusion, *over-parameterized models are capable of reducing noise when trained on noisy data, even without being explicitly tasked to do so.*

We proceed by illustrating the impact of SNR on the test loss curve for both sample and feature noise scenarios in Figures 4(c), 38 respectively. As expected, low SNR values unveil double descent and increase test loss. We then investigate the effect of domain shifts between the training and testing datasets in two cases. First, models are trained on the MNIST dataset and tested on the MNIST-M dataset, as shown in Figure 5(b). Second, models are trained on MNIST-M and tested on MNIST, as seen in Figure 39(b). In both cases, the model-wise double descent curve is observed.

We further illustrate this phenomenon along the epochs axis, displaying non-monotonic behavior and double descent under different levels of sample and feature noise (Figure 40) and showing the impact of SNR variation (Figure 41) for models trained on contaminated MNIST. Additionally, we provide similar results under domain shift conditions between the train and test datasets (Figure 42). Sample-wise double descent and

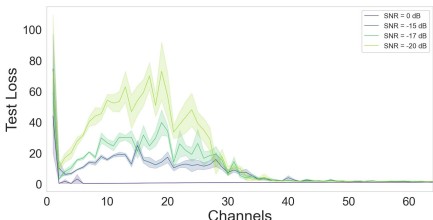 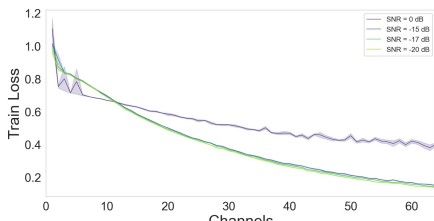

Figure 38: Model-wise double descent for CNN trained on MNIST with varying levels of SNRs and feature noise = 10%.

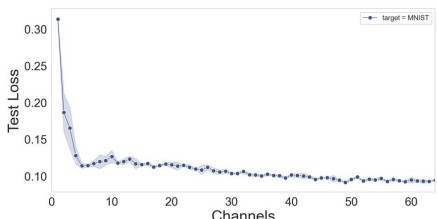 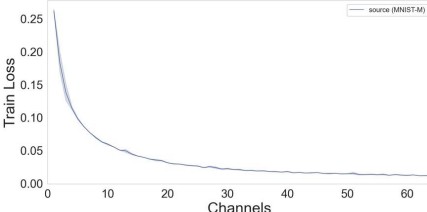

Figure 39: Model-wise double descent for CNNs trained on MNIST-M and tested on MNIST.

non-monotonic behavior is observed as well in all contamination setups. The cases of varying levels of sample noise and feature noise are displayed in Figure 43 and for varying SNRs for both scenarios in Figure 44. Sample-wise double descent is also illustrated in Figure 45 for when a domain shift is present between the training data (MNIST) and the testing data (MNIST-M).

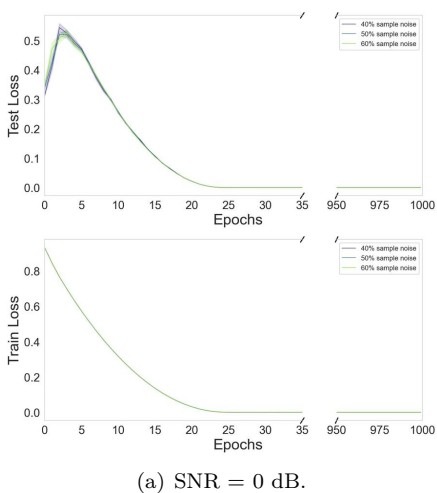 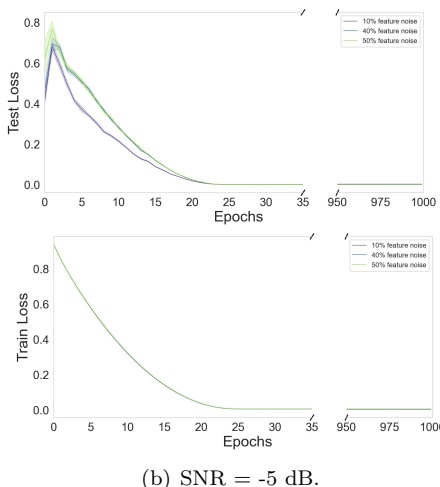

(a) SNR = 0 dB.                                  (b) SNR = -5 dB.

Figure 40: Epoch-wise non-monotonic behavior for CNNs trained on noisy version of MNIST with varying levels of sample noise (a) and feature noise (b).

## F.4 Double Descent Results for the Nonlinear Subspace Data Model

Building on the subspace data model discussed in Subsection 3.1, we have developed a new dataset with nonlinear characteristics to investigate the double descent phenomenon in more complex scenarios. Although the single-cell RNA dataset is already nonlinear, we have created this dataset to demonstrate the reproducibility of the double descent phenomenon across various datasets.

As in the subspace data model discussed in Subsection 3.1, we sample $N$ latent vectors $\{z_i\}_{i=1}^N$ from a normal distribution and project them to a higher dimension using a random matrix $H_1$. The key difference is the inclusion of nonlinear components $z_i^2$ and $z_i^3$, each projected to a higher dimensional space with different

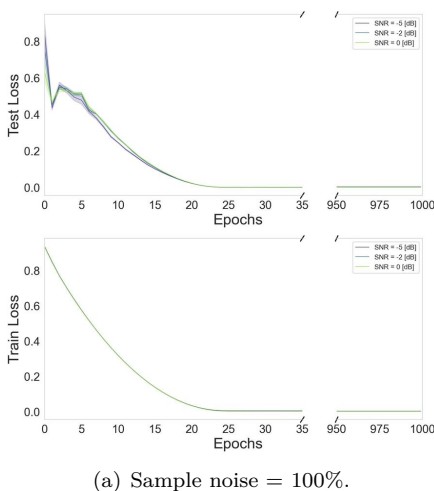 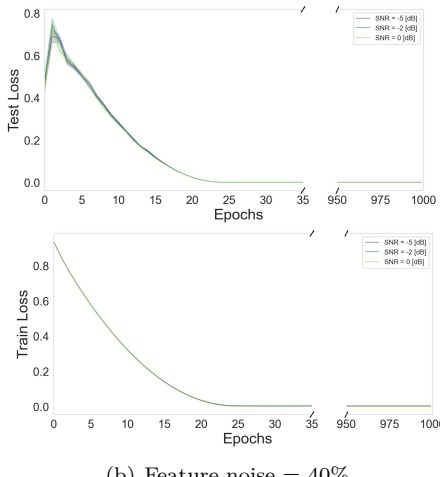

(a) Sample noise = 100%.

(b) Feature noise = 40%.

Figure 41: Epoch-wise double descent and non-monotonic behavior for CNNs trained on MNIST with varying SNRs. (a): sample noise, (b): feature noise.

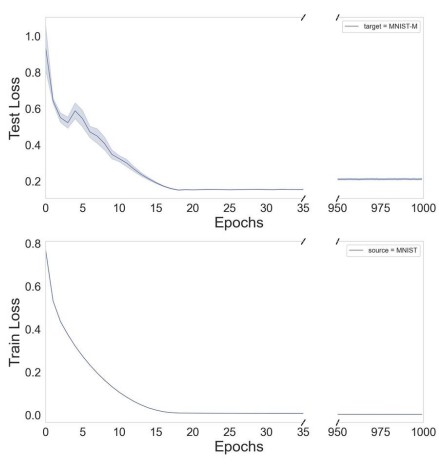 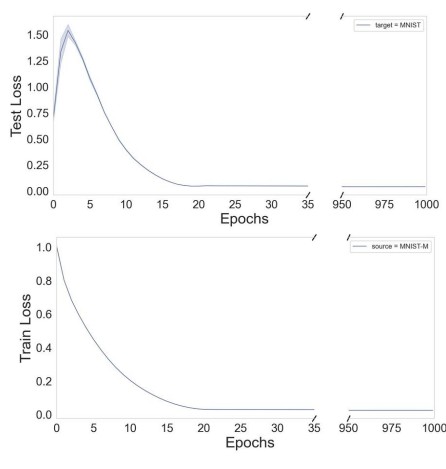

(a) Source = MNIST, target = MNIST-M.

(b) Source = MNIST-M, target = MNIST.

Figure 42: Epoch-wise double descent and non-monotonic behavior for domain shift.

random matrices $\boldsymbol{H}_2$ and $\boldsymbol{H}_3$. To create contaminated setups of sample and feature noise, noise is added to $p \cdot 100\%$ of the data where $\beta$ controls the SNR:

$$
x_i = \begin{cases} \beta(\boldsymbol{H}_1 \boldsymbol{z}_i + \boldsymbol{H}_2 \boldsymbol{z}_i^2 + \boldsymbol{H}_3 \boldsymbol{z}_i^3) + \boldsymbol{\epsilon}_i, & \text{with probability } p, \\ \beta(\boldsymbol{H}_1 \boldsymbol{z}_i + \boldsymbol{H}_2 \boldsymbol{z}_i^2 + \boldsymbol{H}_3 \boldsymbol{z}_i^3), & \text{with probability } 1 - p. \end{cases}
$$

For the domain shift scenario, we divide the latent vectors into training and testing sets and use the same parameter '$s$' as described in Subsection 3.1 to control the shift between the train and test sets in the following manner: $\boldsymbol{H}_j^{''} = \boldsymbol{H}_j + s \cdot \boldsymbol{H}^{'}$ for $1 \le j \le 3$ and get:

$$
\boldsymbol{x}_i = \begin{cases} \boldsymbol{H}_1 \boldsymbol{z}_{train}^i + \boldsymbol{H}_2 (\boldsymbol{z}_{train}^i)^2 + \boldsymbol{H}_3 (\boldsymbol{z}_{train}^i)^3, & \text{if } train, \\ \boldsymbol{H}_1^{''} \boldsymbol{z}_{test}^i + \boldsymbol{H}_2^{''} (\boldsymbol{z}_{test}^i)^2 + \boldsymbol{H}_3^{''} (\boldsymbol{z}_{test}^i)^3, & \text{if } test. \end{cases}
$$

For anomaly detection, clean samples are represented by $\beta(\boldsymbol{H}_1 \boldsymbol{z}_i + \boldsymbol{H}_2 \boldsymbol{z}_i^2 + \boldsymbol{H}_3 \boldsymbol{z}_i^3)$, with $p \cdot 100\%$ of them replaced by anomalies sampled from a normal distribution, as detailed in Subsection 3.1.

The model used in this section is the same FCN model described in Figure 2(a). We start by presenting results for the sample and feature noise scenarios as depicted in Figure 46. As shown, the test loss results for

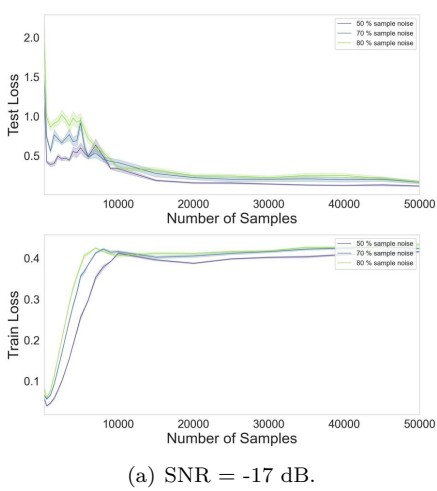
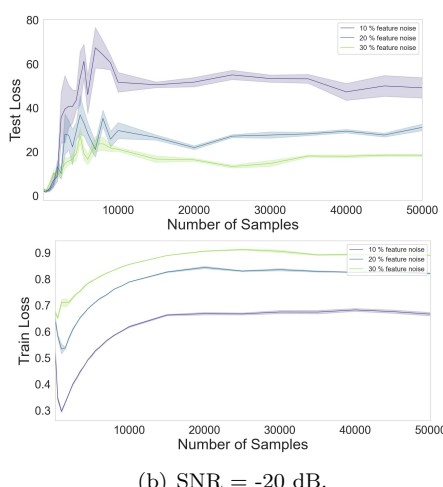

(a) SNR = -17 dB.

(b) SNR = -20 dB.

Figure 43: Sample-wise double descent and non-monotonic behavior for CNNs trained on contaminated MNIST with varying levels of sample noise (a) and feature noise (b).

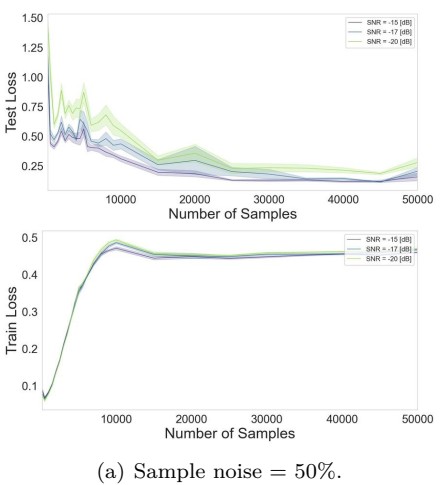
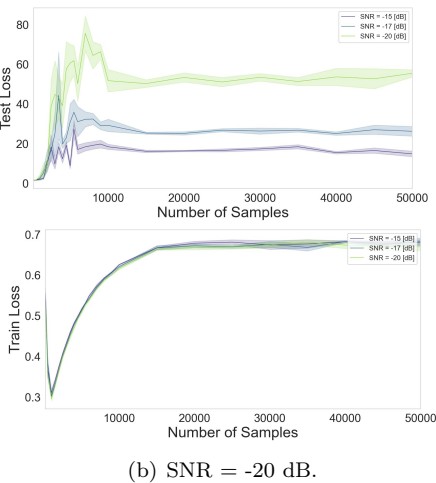

(a) Sample noise = 50%.

(b) SNR = -20 dB.

Figure 44: Sample-wise double descent and non-monotonic behavior for CNNs trained on contaminated MNIST with varying levels of SNR. (a): sample noise, (b): feature noise.

the case of sample noise (Figure 46(a)) resemble those of the subspace data model presented in Figure 3(a). Figure 46(b) demonstrates the model-wise final ascent phenomenon for the case of feature noise as elaborated in Appendix F.5. Figure 47 shows how the SNR affects the test loss curve for both sample and feature noise cases. As observed, the test loss increases with decreasing SNR. Additionally, the final ascent in the test loss is depicted in 47(b) for the feature noise scenario, where the slope becomes steeper as the SNR decreases. We also demonstrate the double descent and final ascent results regarding the domain shift scenario in Figure 48 and the anomaly detection capabilities in Figure 49.

We also observed epoch-wise double descent and non-monotonic behavior for this dataset, as shown in Figure 50 for different percentages of sample and feature noise and in Figure 51 for varying SNRs under the same noise conditions. Additionally, epoch-wise double descent is also observed when a domain shift is present between train and test sets, as depicted in Figure 52. Instances of sample-wise double descent and non-monotonic curves are also reported and displayed in Figure 53 for varying levels of sample and feature noise, Figure 54 for varying levels of SNR, and in Figure 55 for domain shift.

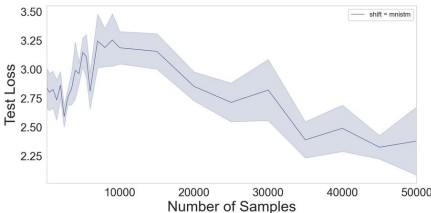 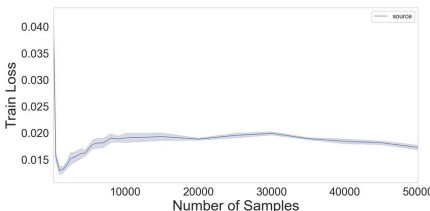

Figure 45: Sample-wise double descent for models trained on the MNIST dataset and tested on the MNIST-M dataset.

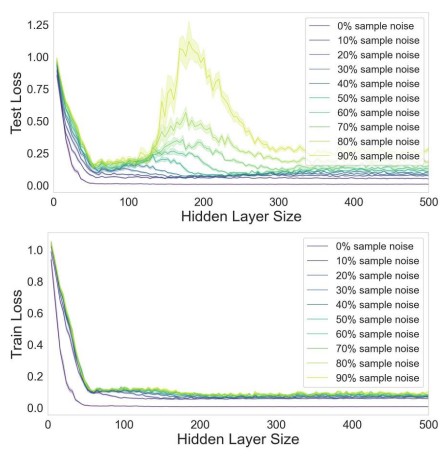 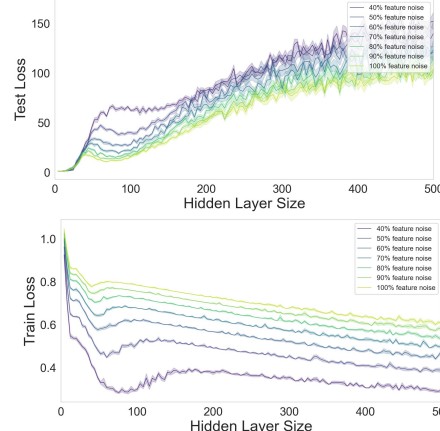

(a) Sample noise scenario with SNR = -15 dB.

(b) Feature noise scenario exhibits final ascent (Appendix F.5) with SNR = -20 dB.

Figure 46: Model-wise double descent for the nonlinear subspace data model with varying levels of sample noise (a) and final ascent with varying levels of feature noise (b).

## F.5 Final Ascent Phenomenon

While training various models on different datasets contaminated with sample and feature noise at different SNR levels and domain shifts between train and test sets, we observed a final ascent phenomenon characterized by a pattern of decreasing-increasing-decreasing-increasing test loss. The phenomenon was first observed in (Xue et al., 2022) in supervised learning with label noise. We suspect a potential connection to this phenomenon in unsupervised learning, which we have yet to fully analyze. We refer to Figure 56(a), which illustrates the final ascent results for the subspace data model under extreme conditions of 100% sample noise, as a continuation of Figure 3(a). We also present the final ascent results for the single-cell RNA dataset in Figure 56(b). Another instance of final ascent with the presence of varying feature noise is illustrated in Figure 27(a) for the subspace data model and in Figures 46(b), 47(b) for the nonlinear subspace data model. Results are also replicated using the nonlinear subspace data model under various domain shifts, as observed in Figure 48.

## F.6 Multiple Descents Under Different Noise Types and Sparse AEs

This section explores the emergence of double and triple descent for noise distributions beyond Gaussian noise and sparse AEs. Figure 57 illustrates the phenomenon for the subspace data model and single-cell RNA datasets when subjected to Laplacian noise. The experimental setup mirrors that of Figure 3(a). As shown, both datasets exhibit similar results under these conditions.

We extend our research to recent applications of AEs, including sparse AEs, which are increasingly utilized in explainable AI (XAI) (Gao et al., 2024) and have been adopted by Google in their Gemini project. Using sparse CNN AEs, we trained models on the MNIST dataset containing 80% noisy samples and observed

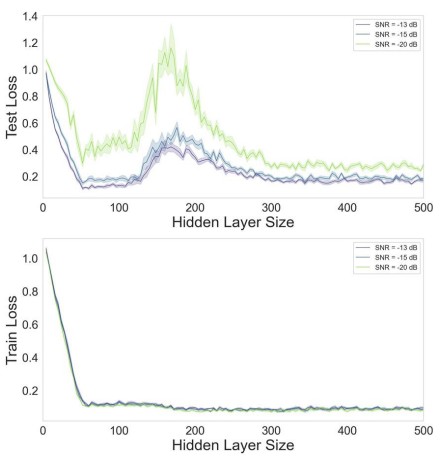
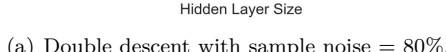
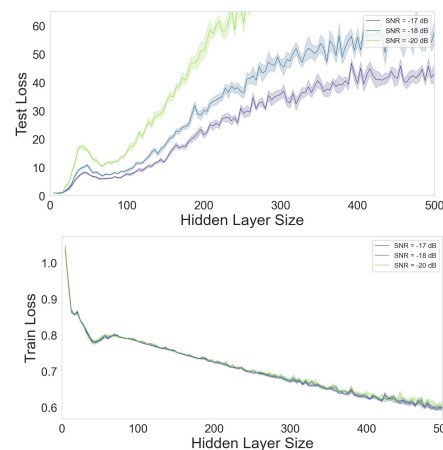
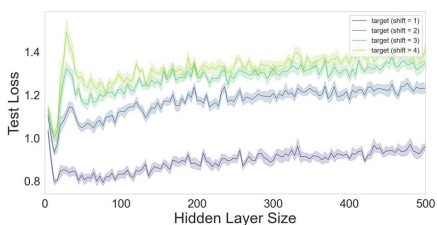

(a) Double descent with sample noise = 80%.

(b) Double descent and final ascent with feature noise = 100%.

Figure 47: Effect of SNR on the test loss curve as a function of model size. (a): **sample noise** scenario. (b): **feature noise** scenario.

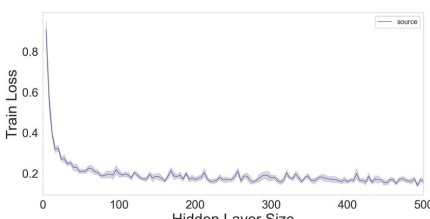

Figure 48: Model-wise double descent and final ascent for the scenario of domain shift.

the emergence of double descent. The models were configured with an embedding layer of size 550, and the parameter $k$, determining the top $k$ highest embedding values to retain, was set to 500. The results are illustrated in Figure 58.

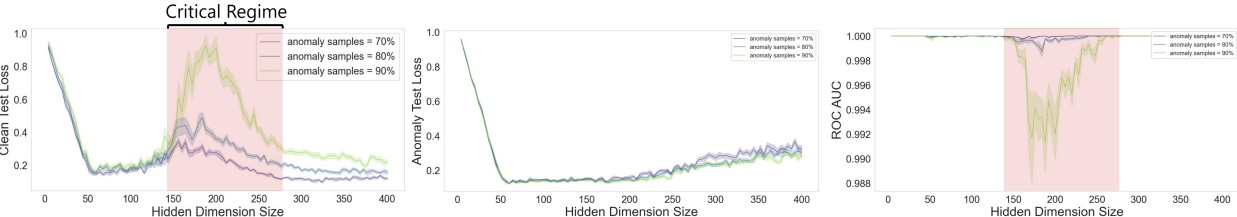

Figure 49: Nonlinear anomaly data with SAR = -15 [dB]. **Left:** test loss of the clean samples. A double descent pattern emerges for low SARs and high anomaly presence in the training data. **Middle:** test loss of the anomaly data. **Right:** Non-monotonic behavior of the ROC-AUC.

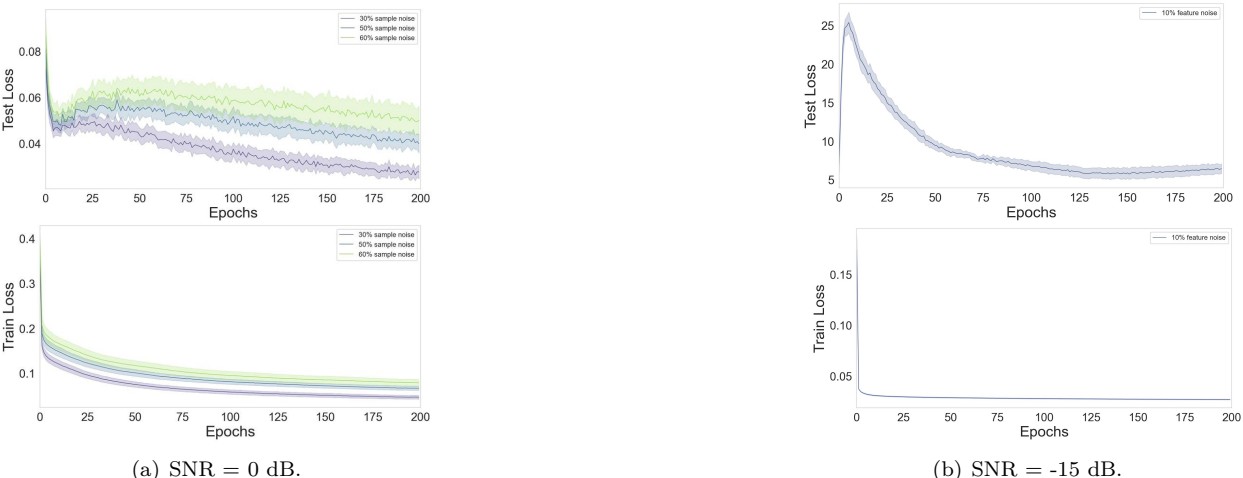

(a) SNR = 0 dB.               (b) SNR = -15 dB.

Figure 50: Epoch-wise double descent and non-monotonic behavior for varying levels of sample noise (a) and feature noise (b). For the scenario of feature noise, we mostly noticed the non-monotonic curve at 10%.

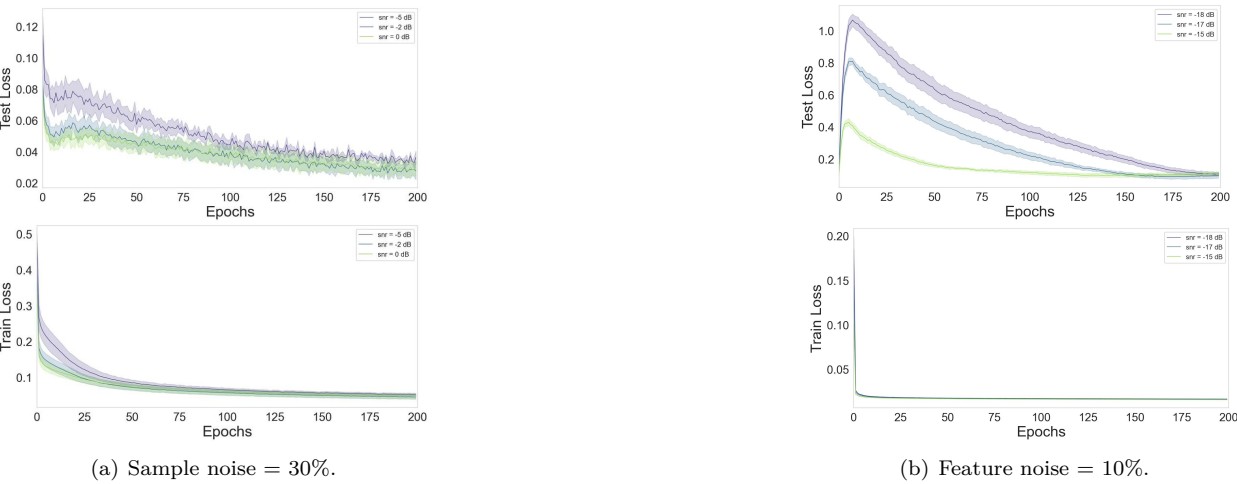

(a) Sample noise = 30%.            (b) Feature noise = 10%.

Figure 51: Epoch-wise double descent and non-monotonic behavior for varying levels of SNR. (a): sample noise, (b): feature noise.

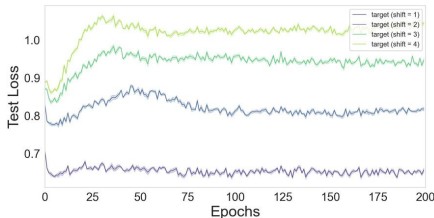 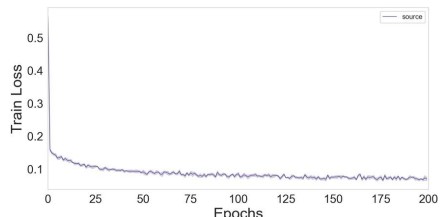

Figure 52: Epoch-wise double descent for when a domain shift is present between train and test sets.

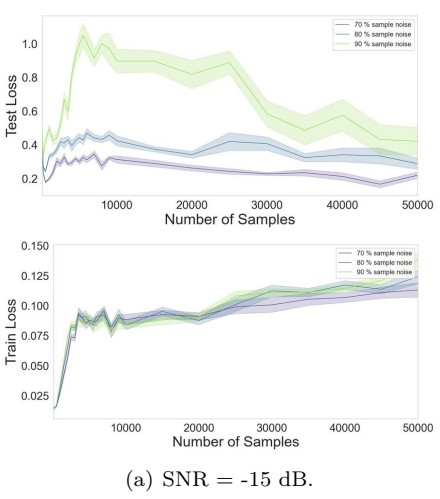 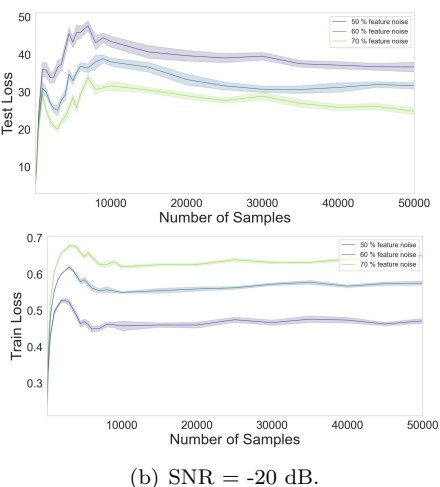

(a) SNR = -15 dB.

(b) SNR = -20 dB.

Figure 53: Sample-wise double descent and non-monotonic behavior for varying levels of sample noise (a) and feature noise (b).

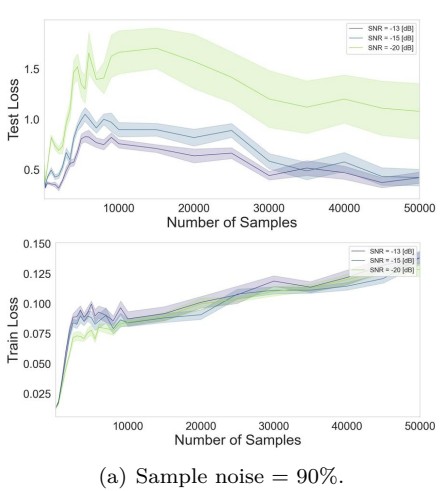 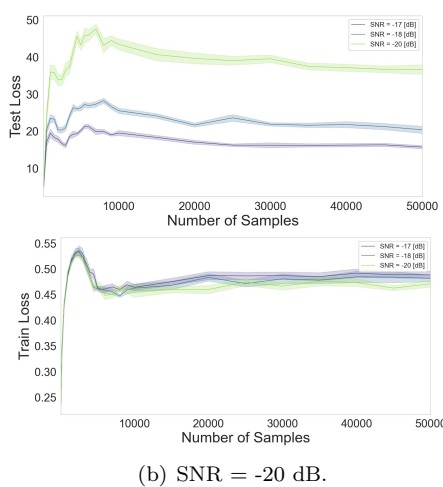

(a) Sample noise = 90%.

(b) SNR = -20 dB.

Figure 54: Sample-wise double descent and non-monotonic behavior for varying levels of SNR. (a): sample noise, (b): feature noise.

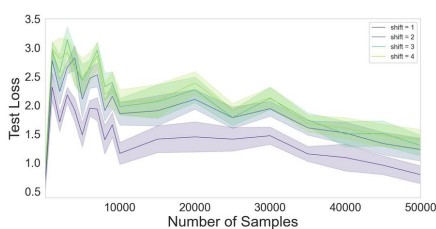 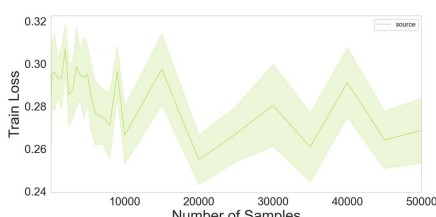

Figure 55: Sample-wise non-monotonic behavior for the domain shift scenario.

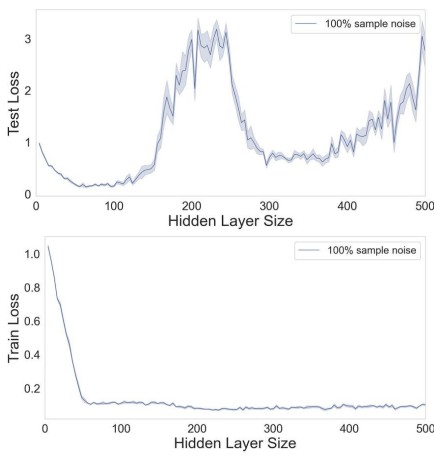

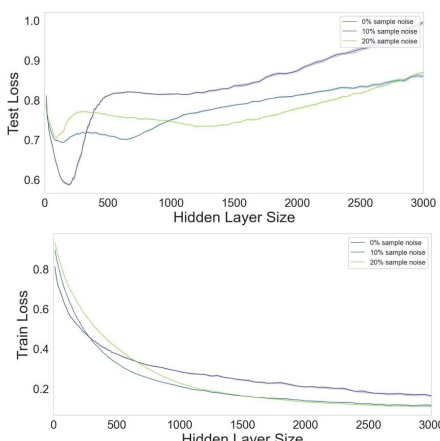

(a) **Subspace data model** and SNR = -15 dB.

(b) **Single-cell RNA data** and SNR = -10 dB.

Figure 56: Test loss exhibits model-wise double descent followed by a final ascent for the scenario of varying **sample noise**.

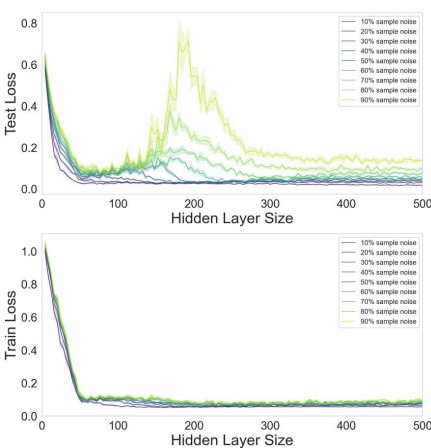

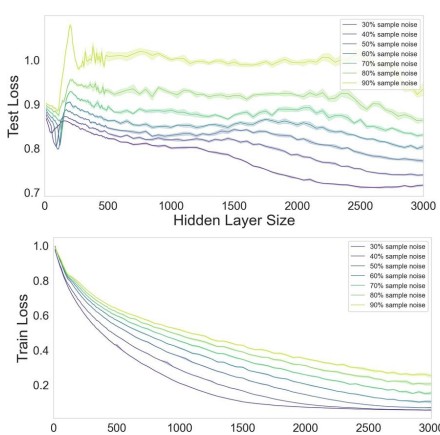

(a) **Subspace data model** and SNR = -15 dB.

(b) **Single-cell RNA data** and SNR = -17 dB.

Figure 57: Test loss exhibits model-wise double descent for the case of Laplace noise.

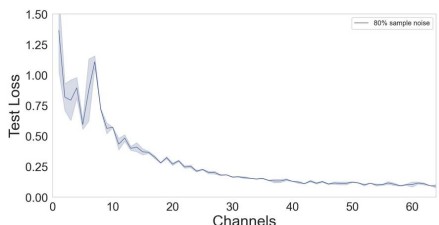

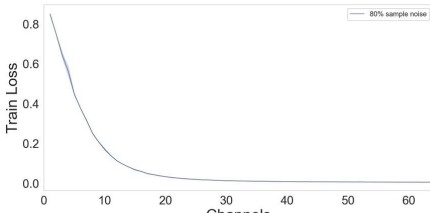

Figure 58: Test loss exhibits model-wise double descent for sparse CNN AEs trained on MNIST with embedding layer size of 550 and k = 500.

