# OpenReview forum: "Unveiling Multiple Descents in Unsupervised Autoencoders"
_TMLR — Accepted by TMLR_

### Review · Reviewer_CF33 · 2025-06-12

**Summary Of Contributions:**

In machine learning, as the model size grows compared to the data size, the test error is expected to decrease and increase as the model becomes overparameterized. Double descent is a surprising phenomenon where further overparameterization leads to a second decrease in test error. Double descent has long been observed in supervised learning, but is relatively understudied for unsupervised learning. The authors attempt to investigate the mystery of double descent in unsupervised learning.

The authors first empirically show that double descent does not happen in linear unsupervised autoencoders, but double (and multiple descent) can consistently present in nonlinear unsupervised autoencoders across model and architectural design. Then, they identify that bottleneck size has a big influence of the double descent curve. They further demonstrate the practical value of the observation by showing that overparameterizing models to the double descent regime can improve downstream tasks.

**Audience:**

Yes

**Broader Impact Concerns:**

Nothing I am aware of.

**Claims And Evidence:**

Yes

**Requested Changes:**

Refer to weakness 1.

**Strengths And Weaknesses:**

Strengths:
1. The paper presents results that challenge the popular belief that double descent is specific to supervised learning. Studying double (and multiple) descent in unsupervised learning is interesting, and the utility studies to connect the observations to real world applications is a plus.
2. The breadth and coverage of the empirical analyses are good. The authors investigated the effect of sample noise, feature noise, domain shift and anomalies, and experimented with two popular types of data with significant domain gap: single-cell RNA sequencing data and CelebA human face images.
3. The observation that the presence of nonlinearities determines the existence of double descent in unsupervised autoencoders is interesting and well supported by results.
4. I like Figure 2(b) and 2(c). These plots provide highly intuitive visualization of double descent along both the hidden layer size axis and the embedding layer size axis in nonlinear autoencoders and the lack thereof in linear autoencoders.

Weaknesses:
1. I would suggest some minor figure-making improvements.

    a. The legends for most double descent figures need to be enlarged, and some legends need to be better placed (for example, Figure 12(b) legend can be put in lower right instead).

    b. The panel placements for most figures are fine, but for Figure 15, enlarging panels (a) through (e) to reduce the whitespace within the same row might be helpful.

    c. [Optional] I personally feel there are too many double descent curves under different conditions but they are not made super obvious --- I understand they are written in captions, but I wonder if centralizing them and making clear distinctions what conditions they respectively correspond to might be helpful. I would recommend the authors consider merging some of them into panels within the same figure, putting the train losses to appendix (like what the authors have already done for some experiments), and clearly labeling the respective conditions. This might make things clearer but it might make things messier, so it’s up to the authors to try and decide the best way of presenting these results.

---

> ### Comment · Reviewer_CF33 · 2025-08-06
> **A minor comment on typo**
>
> Not a big deal:
>
> I see some inconsistency in quotation marks. For example, the double quotes with non-conforming font (for example, "sample noise" in caption of figure 1), the closing quotation marks on both sides of an enclosed phrase in a couple of places (for example, ’Wang’ in caption of figure 9b). The former is likely a copy-paste from other text editor (such as from ChatGPT), and the latter is likely a latex rendering issue: opening quotation marks in latex should be ` instead of '.

---

> > ### Author Response · Authors · 2025-08-06
> >
> > We thank the reviewer for highlighting the inconsistency in quotation marks. We have revised the manuscript accordingly and uploaded an updated version in which all quotation marks have been corrected for consistency.

---

> ### Author Response · Authors · 2025-08-22
>
> Dear Reviewer,
>
> We want to address that we have responded to your review on June 21st, one week after it was posted, and addressed all of your suggestions. The paper has been revised accordingly. We are unsure why the response is not visible to you. In fact, we still see our response posted as an Official Comment to your review. We believe this issue should be addressed by OpenReview or the AE. Below, we paste our original response from June 21st.
>
> We thank the reviewer for the encouraging and constructive feedback, and for acknowledging the strengths of our work, particularly our demonstration of the double descent phenomenon in unsupervised learning, a setting where it remains largely unexplored compared to its well-studied presence in supervised learning. We also appreciate the recognition of our analysis, which connects over-parameterization to improved performance in downstream tasks, as well as our use of real-world datasets that involve anomalies and domain shifts. Finally, we thank the reviewer for highlighting the role of nonlinearity in enabling the phenomenon and for their positive comments on the clarity of Figures 2(b) and 2(c).
>
>
> We believe that we have addressed all of the reviewer’s minor figure improvements in our response below:
>
>
> **Points (a), (b) - legends’ size, adjustment and panel placement**:
> We have increased the size of the legends across all figures in the main paper, including specific adjustments to figures such as the right panels of Figure 9, Figure 12(b), and Figures 15(c–e). Additionally, we have enlarged Figures 15(a–e) to minimize the white space between them and converted the figures to vector graphics to maintain the resolution.
>
>
> **point (c) - merging some figures and clearly labeling the respective conditions**:
> Thank you for this suggestion. We have now combined the test loss results of MNIST with those of the subspace data model and single-cell RNA data into a single figure, while moving the MNIST training loss results to the Appendix. As a result, Figures 3, 4, and 5 now present subfigures from different datasets, illustrating the effects of sample noise, SNR, and domain shift, respectively, eliminating the previous separation of images by datasets trained on FCN AE and CNN AE. The caption of each figure explains which condition is influencing the double descent phenomenon, making it clearer with the results of each dataset combined into a single figure.

---

> > ### Comment · Reviewer_CF33 · 2025-08-25
> > **Response to authors' comment**
> >
> > Dear authors,
> >
> > I believe the same thing happened to me --- I was not able to see your message on June 21st, but luckily I did recognize that you have updated the manuscript and my concerns are resolved.
> >
> > Best

---

### Review · Reviewer_RR8M · 2025-06-14

**Summary Of Contributions:**

The paper investigates multiple descents in unsupervised autoencoders. It provides theoretical insights for linear autoencoders and empirical insights for non-linear autoencoders.

**Audience:**

Yes

**Claims And Evidence:**

Yes

**Requested Changes:**

Below are a few points on how to improve the readability of the paper and make its contribution more appealing to the community.

The acronym "SNR" is not explained but used quite frequently -- it is clarified in Appendix B, though. However, to be more self-contained, I recommend explaining it when used for the first time.

Is there any reason behind using the single-cell RNA data set and the CelebA data set? I think, in particular, for the real-world data sets, it is important to justify why they have been selected.
Furthermore, I am missing at least a brief description of the data sets. It is okay to put the details in the appendix, but in order to be more self-contained, the paper should provide a minimal level of information, such as the domain and dimensionality of the data.
I think it might be a good idea to have a section briefly describing all the data sets used in this paper. For instance, after describing those initial data sets, on page 6 (Figure 5), you suddenly present an experiment involving MNIST which was not mentioned before. Later on, you also have experiments on CIFAR-10 -- please also provide a reference for those data sets.

Sections 3.2 - 3.5 are breaking the flow. Can you try to restructure them such that there is a better reading flow?

Section 5: What is "undercomplete FCN"?

Conclusion: I miss a discussion of the consequences or impact of this work. What are the lessons learned? Are there any research questions that are still open or are arising after your study?


Minor:
- page 5 "following sentence" -> "following theorem"
- The generalization error (GE) is denoted as G. However, in the text, GE is used instead of G. This might be a bit confusing. Maybe you can make it more consistent.
- Whenever possible, use vector graphics

**Strengths And Weaknesses:**

# Strengths
The paper investigates an open research topic and provides novel theoretical and empirical insights that should be of interest to the community.

# Weaknesses
Readability can be improved, outlining the consequences of this work and highlighting next steps for future research.

---

> ### Comment · Reviewer_RR8M · 2025-08-21
>
> Dear authors,
> I might have missed it, but have you already responded to my review? I can not see any response here (to my review).
> Please do so as soon as possible.
> Best regards,
> Reviewer RR8M

---

> > ### Author Response · Authors · 2025-08-22
> >
> > Dear Reviewer,
> >
> > We responded to your review on June 21st, one week after it was posted, and addressed all of your suggestions. The paper has been revised accordingly. We are unsure why the response is not visible to you. Reviewer CF33 has responded to our Official Comment, which was posted on the exact same date. In fact, we still see our response posted as an Official Comment to your review. We believe this issue should be addressed by OpenReview or the AE. Below, we paste our original response from June 21st.
> >
> > We thank the reviewer for their thoughtful and encouraging feedback. We appreciate the recognition of our work's contribution to understanding multiple descent phenomena in unsupervised autoencoders, both from theoretical and empirical perspectives. We are glad that the novelty and relevance of our findings were acknowledged and that the work is considered a valuable addition to the community.
> >
> > We believe that we have addressed all of the reviewer’s requested changes in our response below:
> >
> > **SNR:**
> >
> > We appreciate the reviewer’s suggestion to clarify the acronym 'SNR' for better self-containment. We have added an explanation upon its first mention in the paper, including the rationale behind it and the corresponding formula. We then refer the reader to Appendix B for further elaboration. This addition appears in Subsection 3.1 on page 3.
> >
> > **Justification of the chosen datasets and a brief description of the datasets :**
> >
> > We appreciate the reviewer’s suggestions for justifying the rationale behind the dataset selection and improving the clarity and self-containment of the paper. In response, we revised Subsections 3.2 and 3.3, and included a new discussion on the nonlinear data model at the end of Subsection 3.1 (end of page 3), which is further elaborated in the Appendix. Additionally, we introduced Subsection 3.4 to describe the MNIST and CIFAR-10 datasets, bringing the total number of datasets used to six.
> >
> > Our paper presents real-world, high-dimensional datasets from diverse domains, trained on different architectures. This setup allows us to demonstrate the generalization of the double descent phenomenon across both datasets and model types. Due to their high dimensionality, these datasets are well-suited for dimensionality reduction via AEs, as also explored in prior work [1, 2, 3].
> >
> > Specifically:
> >
> > We used the single-cell RNA dataset due to its inherent domain shifts, which include 5 distinct domains (biological batches) stemming from differences in laboratory conditions and measurement technologies. This makes it particularly suitable for testing our claims about double descent under real-world distribution shifts, which are common in biological and medical data. Each batch represents 15 different cell types, where each cell (sample) in this dataset contains over 15,000 genes (features), making it a high-dimensional dataset.
> >
> > The celebA dataset was selected to evaluate the impact of model complexity on unsupervised anomaly detection performance. CelebA provides a rich, real-world setting with data attributes, making it particularly suitable for testing whether model complexity translates into non-monotonic trends in downstream tasks such as anomaly detection.
> >
> > MNIST and CIFAR-10 are now also mentioned in the main paper and were employed to evaluate our findings on standard image benchmarks, demonstrating double descent under sample noise, feature noise, and domain shift scenarios. These datasets also support reproducibility across different model architectures and data domains.
> >
> > **"Sections 3.2 - 3.5 are breaking the flow":**
> >
> > We have reorganized the content by separating the datasets and model architectures into two distinct sections: Section 3 now focuses on the datasets used in our experiments, while Section 4 describes the various model architectures. We hope this restructuring improves the overall reading flow. Please let us know if you have alternative suggestions—we are committed to making the paper as clear and accessible as possible.
> >
> > **Section 5:**
> >
> > ‘What is "undercomplete FCN"?’ - We thank the reviewer for pointing out this oversight. Our intention was to refer to undercomplete AEs implemented using FCNs. We have revised the phrasing to 'FCN undercomplete AEs' for clarity.
> >
> > **Conclusion:**
> >
> > We have revised parts of the conclusion to explicitly discuss the consequences and impact of our work, namely, the extension of double descent theory to unsupervised learning, its practical relevance for real-world tasks, and its implications for model design. In addition, we have included a discussion of open research questions that arise from our findings. These include exploring how double descent manifests in other unsupervised learning frameworks and investigating how varying levels of nonlinearity, such as different activation functions ranging from linear to highly nonlinear affect the shape and presence of the double descent curve.

---

> > > ### Author Response · Authors · 2025-08-22
> > >
> > > **Minor:**
> > >
> > > * Page 5 - rephrased from "following sentence"  to "following theorem".
> > > * The generalization error is now represented by ‘g’ throughout the paper.
> > > * All figures in the main paper were converted to vector graphics.
> > >
> > > [1] Gökcen Eraslan, Lukas M Simon, Maria Mircea, Nikola S Mueller, and Fabian J Theis. Single-cell rna-seq denoising using a deep count autoencoder. Nature communications, 10(1):390, 2019.
> > >
> > > [2] Yasi Wang, Hongxun Yao, and Sicheng Zhao. Auto-encoder based dimensionality reduction. Neurocomputing, 184:232–242, 2016.
> > >
> > > [3] Qinxue Meng, Daniel Catchpoole, David Skillicom, and Paul J Kennedy. Relational autoencoder for feature extraction. In 2017 International joint conference on neural networks (IJCNN), pp. 364–371. IEEE, 2017.

---

> > > > ### Comment · Reviewer_RR8M · 2025-08-22
> > > >
> > > > Thanks for your response and clarifications. These resolve all my concerns.

---

### Review · Reviewer_p8vq · 2025-07-22

**Summary Of Contributions:**

The authors study the double descent phenomenon in what is perhaps the most basic unsupervised model that can be overparametrised: autoencoders (AEs).

They start from the simplest kind of AE: linear ones. In this context, they show (empirically and theoretically) that double descent does not occur. As I detail below, I have some issues with this claim.

Then, they move on to nonlinear AEs, and show empirically that double descent *does* occur in this context, both in terms of the dimension of the latent space and in terms of the size of the involved networks.

All experiments are performed over a wide variety of datasets, architectures, and corruption schemes (double descent is often easier to spot in setting with corrupted data). They also look at the impact of double descent on downstream tasks (domain adaptation and anomaly detection).

**Audience:**

Yes

**Broader Impact Concerns:**

I have no concerns here.

**Claims And Evidence:**

No

**Requested Changes:**

Clarifying whether or not the main theorem is true, and if it implies that PCA does not overfit with respect to the latent space dimension.

**Strengths And Weaknesses:**

**Strengths**

Double descent in unsupervised settings remains a wide open question, and it is refreshingly interesting to see a paper that tackes this topic.

The authors do a very good job at motivating their study, and the related works section is quite well-written. I am not an expert on double descent so I may have missed some papers, but to the best of my knowledge the authors are right to claim that  "this is the first demonstration of model and epoch, and sample-wise double descent in a fully unsupervised setting". This is an important contribution!

I thought the experiments were nicely designed, and I enjoyed the focus on downstream tasks.

**Weaknesses**

I have one major concern about this paper: I do not follow the reasoning of the proof of the main theorem. The main theorem claims that linear AEs with a larger latent space will *always* generalise better (the only regularity condition being that the covariance matrix of the data is full rank almost everywhere). In other words, linear AEs do not overfit with respect to the latent space dimension.

First, I find the theorem quite hard to believe from practical grounds: unless I am missing something, if it were true, when doing dimensionality selection in PCA using a validation set, the model with the largest dimensionality would always be chosen. Choosing the number of latent dimensions in PCA using a validation set is very commonly done (see e.g. Minka, 2000, Josse & Husson 2012), and generally works very well. However, this theorem seems to imply that this is impossible.

Regarding the proof (available in Appendix C), I do not follow a particular point. The authors denote the empirical covariance of the data set by $\Sigma = X X^T$, whose $m$ largest eigenvectors are stored in a matrix $U_m$, and the theoretical covariance of the test distribution by $\tilde{\Sigma} = \mathbb{E} [ x x^T]$, whose $m$ largest eigenvectors are stored in a matrix $\tilde{U}_m$. They then claim that there exists an unitary matrix $R$ such that $U_m = R \tilde{U}_m$. I do not see how this can be the case, since $U_m$ and $\tilde{U}_m$ have no reason to span the same subspace. I also do not understand what is the dimension of this matrix $R$. In some lines it looks like it should be $D$ by $D$, and is others $m$ by $m$.



**Minor things**

- In many cases in the experiments, the distribution of the test set is not the same as the distribution of the training set. It would be nice to summarise somewhere which experiments are following the standard ML dogma of using the same distribution, and which experiments are not.

- This is very minor, but the common practice in ML is generally to write data matrices in a way such that each row is an observation, and each column a feature. Here the authors do the opposite.

**Additional references**

- Josse & Husson, Selecting the number of components in principal component analysis using cross-validation approximations, CSDA 2012
- Minka, Automatic choice of dimensionality for PCA, NeurIPS 2000

---

> ### Author Response · Authors · 2025-07-27
> **Response to reviewer p8vq**
>
> We sincerely thank the reviewer for their thoughtful and encouraging feedback. We are particularly grateful for the recognition of the novelty of our work and its contribution to the study of double descent in fully unsupervised settings. It is encouraging to hear that our motivation and related work were clearly conveyed and appreciated.
>
> We also appreciate the positive remarks regarding our experimental design and the emphasis on downstream evaluation, which we believe provides an important practical perspective.
>
> We provide additional details and address the reviewer’s concerns below.
>
> We thank the reviewer for showing interest in the reasoning of our proof and for referencing relevant prior work on dimensionality selection in PCA.
>
> To clarify our theoretical framework and its motivation, we aim to analyze the generalization behavior of unsupervised models, specifically autoencoders (AEs), in a context similar to that used in double descent studies for supervised learning. In these studies, models are often trained on noisy data (with label noise) and evaluated on clean data, allowing researchers to assess whether the model learns the actual structure of the data or merely memorizes the noise. Our approach similarly allows us to examine the impact of partial noise in the training distribution. We seek to understand how learning from corrupted data affects the model’s ability to capture the underlying clean structure.
>
> We consider a scenario where the AE is trained on samples from a noisy input distribution $P_x$, but evaluated for reconstruction accuracy against a clean distribution $\tilde{P_x}$, where $P_x \neq \tilde{P_x}$ (elaborated in the revised manuscript in blue color, sections 5, 6, and 7) . In this context, the model's expected reconstruction error on clean inputs, despite having been trained on noisy examples, indicates generalization. This is the definition of generalization error, G, used in our theoretical study in Section 5. By using undercomplete linear AEs, we ensure that the model cannot learn the identity map, allowing us to study the meaningful impact of the latent dimension on the model's ability to denoise and generalize.
>
> Our main theoretical result (Theorem 5.2) demonstrates that, under mild assumptions (e.g., full-rank covariance and no explicit regularization), increasing the latent dimension does not degrade generalization performance in this setting. This result helps isolate the role of model capacity in denoising and contrasts with classical intuition about overfitting. Importantly, we note that **if the test data were also noisy**, our result does not contradict the reviewer's intuition, and indeed, it could be beneficial to remove some coordinates to attenuate noise components.
>
> Our theoretical result differs from the objectives and assumptions of the works by Minka (2000), Josse & Husson (2012), and similar approaches based on principles like the Marchenko–Pastur law [1, 2]. These methods focus on distinguishing signal from noise in both training and test data by reducing the influence of components that are primarily noise. This is typically based on the assumption that both training and test samples are drawn from the same noisy distribution.
>
> We acknowledge that dimensionality selection remains crucial in practice, particularly when working with noisy test data. However, our theoretical findings indicate that having partial noise in the training data does not negatively impact generalization to clean data in an idealized linear AE setting. In this scenario, increasing the latent dimension does not lead to overfitting, which suggests that the phenomenon of double descent is absent. This distinction between idealized and practical contexts motivates our broader empirical investigation into how factors such as nonlinearity, data contamination, and architectural choices contribute to the occurrence of the double descent phenomenon in nonlinear AEs.
>
> [1] Gavish, Matan, and David L. Donoho. "Optimal shrinkage of singular values." IEEE Transactions on Information Theory 63.4 (2017): 2137-2152.
>
> [2] Butsch, Lucas, and Vicky Fasen-Hartmann. "Estimation of the number of principal components in high-dimensional multivariate extremes." arXiv preprint arXiv:2505.22437 (2025).

---

> > ### Author Response · Authors · 2025-07-27
> >
> > Regarding the proof, we thank the reviewer for this helpful comment. We would like to clarify the point regarding the dimensions of the matrices in the proof, as we believe part of the confusion stems from a distinction between the **full eigenbasis matrices** and their **truncated forms**.
> >
> > Specifically, $\boldsymbol{U},\tilde{\boldsymbol{U}}\in\mathbb{R}^{D\times D}$ denote the full eigenvector matrices of the empirical and theoretical covariance matrices, respectively, while $\boldsymbol{U}_m,\tilde{\boldsymbol{U}}_m\in\mathbb{R}^{D\times m}$ represent the truncated matrices containing the top $m$ eigenvectors.
> > The rotation matrix $\boldsymbol{R}\in\mathbb{R}^{D\times D}$ aligns the full eigenbases, satisfying $\boldsymbol{U}= \boldsymbol{R}\tilde{\boldsymbol{U}}$. This matrix is constructed as $\boldsymbol{R}=\boldsymbol{U}\tilde{\boldsymbol{U}}^{\top}$ and is orthonormal by definition, i.e., $\boldsymbol{R}^\top \boldsymbol{R}=\mathbb{I}_D$.
> > While it is true that the truncated eigenvector matrices $\boldsymbol{U}_m$ and $\tilde{\boldsymbol{U}}_m$
> >
> > generally span different $m$-dimensional subspaces, both reside within the same ambient $D$-dimensional space, thus enabling such an orthonormal transformation $\boldsymbol{R}$.
> > Note that since both, $\boldsymbol{U}^{\textcolor{black}{D \times D}}$ and $\tilde{\boldsymbol{U}}^{\textcolor{black}{D \times D}}$ are orthonormal and span $\mathbb{R}^D$ (by the SVD property), there exists an orthogonal matrix $\boldsymbol{R}^{D \times D}$ such that $\boldsymbol{U}= \boldsymbol{R}\tilde{\boldsymbol{U}}$. This matrix $\boldsymbol{R}= \boldsymbol{U}\tilde{\boldsymbol{U}}^\top$ is unitary by construction, satisfying $\boldsymbol{R}^\top \boldsymbol{R}= \mathbb{I}_{D \times D}$, and represents a rotation or reflection aligning the test and training eigenspaces. Thus $\boldsymbol{U}$ and $\tilde{\boldsymbol{U}}$ are related by this $\boldsymbol{R}$, such that $
> > \boldsymbol{U}=\boldsymbol{R}\tilde{\boldsymbol{U}}.
> > $
> >
> >
> > To avoid ambiguity, we have revised the manuscript (Section 5 and Appendix C) to clearly distinguish between full and truncated eigenbases and to explicitly state the dimensions of all relevant matrices. We hope this clarifies the structure of the argument and eliminates potential confusion regarding the role and dimensions of $\boldsymbol{R}$.
> >
> > **Minor Things:**
> >
> > With respect to the use of similar distributions for training and testing in the ‘minor things’ section, the only experiments where this applies are those related to **anomaly detection**. In these experiments, models are trained on contaminated data containing both clean and anomalous samples, and are evaluated on a test set that also includes clean and anomalous data, enabling us to compute the ROC curve and its corresponding AUC. This setup is illustrated in Figures 6 and 13, where the left panels show the test loss for clean samples and the middle panels show the loss for anomalous ones. We have also added a clarification in section 6, ‘Anomalies’ section (page 7), noting that the training and testing distributions are aligned.

---

> > > ### Comment · Reviewer_p8vq · 2025-08-02
> > >
> > > Thanks for your detailed response and the update of the manuscript. I will respond to all points after we have dealt with the proof.
> > >
> > > Thanks to your clarifications, I think I get the manipulations of Equation (4), although I think including an additional step between the third and the fourth line would be helpful (I guess you use the fact that $I- U_m U_m^T$ is idempotent?). However, I am still puzzled by the result in general. You say that the result would not be true if the test were also noisy: where does the "cleanliness" of the test set appear in the proof?

---

> > > > ### Author Response · Authors · 2025-08-04
> > > >
> > > > We thank the reviewer for their thoughtful and constructive feedback.
> > > > Regarding the manipulations in Equation (4), we appreciate the suggestion to include an additional intermediate step for clarity. We have added new lines between the third and fourth lines in the revised manuscript to explicitly highlight the use of the identity  $ I -  U_{m} U_{m}^{T}​$ being idempotent. This addition indeed helps clarify the logical flow of the proof and improves its readability for the reader.
> > > >
> > > > We appreciate the reviewer’s insightful question. In the paper we trained models on contaminated datasets, drawn from $P_x$ and tested on clean data sampled from $P_{\tilde{x}}$ to isolate the effect on the learned model, showing noise memorization (high test loss) versus signal learning (low test loss) and observed model, epoch, and sample-wise multiple descents in non-linear models. We followed the same approach as in supervised learning, where models are trained with noisy labels and tested on clean ones (in our unsupervised scenario, trained on noisy data and tested on clean data). During testing, we evaluate clean-to-clean reconstruction using samples drawn from $P_\tilde{x}$ (this has also been done in [1]) in contrast to what is often done in denoising setups where the input is noisy and the target is clean. This distinction is captured in the definition of the generalization loss:
> > > > $$\mathcal{G}(\mathcal{D};\boldsymbol{\theta}^{opt},P_{\tilde{x}})\=\mathbb{E}\left[\| |\Phi(\tilde{x};\boldsymbol{\theta}^{opt})
> > > > -\tilde{x}\||_2^2\right].$$
> > > >
> > > > To clarify how the "cleanliness" of samples impacts our proof, when noise is added to the test set, $\tilde{x}$ becomes $\tilde{x} + n$, which introduces a noise term in the generalization error expression. This modification influences the analysis in the following way:
> > > >
> > > > \begin{align*}
> > > > \mathcal{G}^\prime(D; \boldsymbol{\theta}^{{opt}}, P_x)
> > > > = \mathbb{E}\Big[ \big\|\big| \Phi(\tilde{x}+n; \theta^{\text{opt}}) - \tilde{x} \big\|\big\|_2^2 \Big] \\
> > > > = \mathbb{E}\Big[ \big\|\big| \mathbf{U}_m \mathbf{U}_m^\top (\tilde{x}+n) - \tilde{x} \big\|\big|_2^2 \Big] =\\
> > > > \end{align*}
> > > > \begin{align*}
> > > > = \mathbb{E}\Big[ \big\|\big\| (\mathbf{U}_m \mathbf{U}_m^\top - \mathbb{I}) \tilde{x} + \mathbf{U}_m \mathbf{U}_m^\top n \big\|\big\|_2^2 \Big] \\
> > > > = \mathbb{E}\Big[ \big\|\big\| (\mathbb{I} - \mathbf{U}_m \mathbf{U}_m^\top) \tilde{x} \big\|\big\|_2^2 + \big\|\big\| \mathbf{U}_m \mathbf{U}_m^\top n \big\|\big\|_2^2 + 2 \big\langle (\mathbb{I} - \mathbf{U}_m \mathbf{U}_m^\top) \tilde{x}, \mathbf{U}_m \mathbf{U}_m^\top n \big\rangle \Big] \\
> > > > \end{align*}
> > > > \begin{align*}
> > > > = \mathbb{E}\Big[ \big\|\big\| (\mathbb{I} - \mathbf{U}_m \mathbf{U}_m^\top) \tilde{x} \big\|\big\|_2^2 \Big] + \mathbb{E}\Big[ \big\|\big\| \mathbf{U}_m \mathbf{U}_m^\top n \big\|\big\|_2^2 \Big] \\
> > > > = \mathrm{Tr}\Big[ (\mathbb{I} - \mathbf{U}_m \mathbf{U}_m^\top) \tilde{\Sigma_x} \Big] + \mathrm{Tr}\Big[ \mathbf{U}_m \mathbf{U}_m^\top \Sigma_n \Big] \\
> > > > = \mathrm{Tr}\Big[ (\mathbb{I} - \mathbf{U}_m \mathbf{U}_m^\top) \tilde{\Sigma_x} \Big] + \mathrm{Tr}\Big[ \mathbf{U}_m \mathbf{U}_m^\top \Sigma_n \Big].
> > > > \end{align*}
> > > >
> > > > The left term is the clean error, and the right term represents the projected noise contribution.  We used the assumption that the expectation of the noise is zero. When $m$ increases, the first term decreases while the second term increases, implying that increasing $m$ does not have to monotonically decrease this definition of the generalization error, which is in line with prior work [2]. When noise is added to the test set, the mathematical results suggest a ``U-shaped’’ trend in the test loss. This aligns with the analysis in the referenced papers and your intuition about the importance of choosing a balanced number of components.
> > > > This theoretical analysis can also be found in the revised manuscript in Remark C.3 (pages 23–24).
> > > >
> > > > We hope this clarifies how the "cleanliness" of the test set takes part in the formulation and the interpretation of our results.
> > > >
> > > > [1] Lupidi, Alisia, Yonatan Gideoni, and Dulhan Jayalath. "Does Double Descent Occur in Self-Supervised Learning?." arXiv preprint arXiv:2307.07872 (2023).
> > > >
> > > > [2] Gavish, Matan, and David L. Donoho. "Optimal shrinkage of singular values." IEEE Transactions on Information Theory 63.4 (2017): 2137-2152.

---

> > > > > ### Comment · Reviewer_p8vq · 2025-08-06
> > > > >
> > > > > Thanks for your response!
> > > > >
> > > > > Indeed, Equation (4) is clearer (at least to me). I would also add an explicit mention that you use the idempotence of the matrix (and why it is idempotent), but this is nitpicking.
> > > > >
> > > > > There is one thing that bothers me about your alternative definition of the generalisation error:
> > > > >
> > > > > \begin{align*} \mathcal{G}^\prime(D; \boldsymbol{\theta}^{{opt}}, P_x) = \mathbb{E}\Big[ \big|\big| \Phi(\tilde{x}+n; \theta^{\text{opt}}) - \tilde{x} \big|\big|_2^2 \Big],\end{align*}
> > > > >
> > > > > which is that measures how well the clean data will be reconstructed from the noisy one. This is not the standard generalisation error looked at in PCA or matrix factorisation, because it is impossible to compute using a test/validation set in practice (because we would need the clean data $\tilde{x}$ to do so). A more natural (and closer to statistical practice) measure would be to juste compute the average generalisation MSE, that would be, in your notations
> > > > >
> > > > > \begin{align*} \mathcal{G}_2^\prime(D; \boldsymbol{\theta}^{{opt}}, P_x) = \mathbb{E}\Big[ \big|\big| \Phi(\tilde{x}+n; \theta^{\text{opt}}) - (\tilde{x} + n) \big|\big|_2^2 \Big].\end{align*}
> > > > >
> > > > > What happens in that case? This should add another term that depends both on the distribution of the noise and the latent dimension.

---

> > > > > > ### Author Response · Authors · 2025-08-07
> > > > > >
> > > > > > Thank you for your suggestion. To clarify, in response to your earlier request to examine the impact of “cleanliness” on our analysis, we added noise to the test samples and evaluated how well the Autoencoder (AE) reconstructs the clean signal. This approach is consistent with classical signal recovery frameworks, such as optimal shrinkage in PCA, where our goal is to assess the recovery of the underlying clean signal from noisy observations [1, 2, 3].
> > > > > >
> > > > > > Even in the absence of clean samples, techniques like Marchenko-Pastur theory [4] can estimate noise levels and help differentiate between signal and noise. When possible, evaluating against clean targets offers clearer insights into generalization, especially in overparameterized scenarios where memorization can occur.
> > > > > >
> > > > > > On the other hand, analyzing the reconstruction of noisy inputs is a different type of analysis and is less directly relevant to signal recovery, as perfect reconstruction would retain the noise. Nonetheless, to complete the analysis, we provide the derivation below in response to your request:
> > > > > >
> > > > > > \begin{align*}
> > > > > > \mathcal{G}^{\prime}_{2} (D;\boldsymbol{\theta}^{opt},\tilde{P_x})
> > > > > > = \mathbb{E}\Big[ \big\|\big\| \Phi(\tilde{x}+n; \theta^{\text{opt}}) - (\tilde{x}+n) \big\|\big\|^2_2 \Big] \\
> > > > > > = \mathbb{E}\Big[ \big\|\big\| (\mathbb{I} - \mathbf{U}_m \mathbf{U}_m^\top) (\tilde{x}+n) \big\|\big\|^2_2 \Big] =\\
> > > > > > \end{align*}
> > > > > > \begin{align*}
> > > > > > = \mathbb{E}\Big[ (\tilde{x}+n)^\top (\mathbb{I} - \mathbf{U}_m \mathbf{U}_m^\top)^\top (\mathbb{I} - \mathbf{U}_m \mathbf{U}_m^\top) (\tilde{x}+n) \Big] \\
> > > > > > = \mathbb{E}\Big[ (\tilde{x}+n)^\top (\mathbb{I} - \mathbf{U}_m \mathbf{U}_m^\top) (\tilde{x}+n) \Big] =\\
> > > > > > \end{align*}
> > > > > > \begin{align*}
> > > > > > = \mathbb{E}\Big[ \mathrm{Tr}\big( (\mathbb{I} - \mathbf{U}_m \mathbf{U}_m^\top) (\tilde{x}+n)(\tilde{x}+n)^\top \big) \Big] \\
> > > > > > = \mathrm{Tr}\Big[ (\mathbb{I} - \mathbf{U}_m \mathbf{U}_m^\top) \ \mathbb{E}\big[ (\tilde{x}+n)(\tilde{x}+n)^\top \big] \Big] =\\
> > > > > > \end{align*}
> > > > > > \begin{align*}
> > > > > > = \mathrm{Tr}\Big[ (\mathbb{I} - \mathbf{U}_m \mathbf{U}_m^\top) (\Sigma_x + \Sigma_n) \Big] \\
> > > > > > = \mathrm{Tr}\Big[ (\mathbb{I} - \mathbf{U}_m \mathbf{U}_m^\top) \Sigma_x \Big] + \mathrm{Tr}\Big[ (\mathbb{I} - \mathbf{U}_m \mathbf{U}_m^\top) \Sigma_n \Big]=\\
> > > > > > \end{align*}
> > > > > >
> > > > > > \begin{align*}
> > > > > > =  \mathrm{Tr}(\boldsymbol{\Sigma}_{\tilde{x}}) -\mathbb{E}\left[\big\|\big\|\mathbf{\tilde{U}}_m^{\top} \mathbf{R}^{\top} \tilde{x}\big\|\big\|_2^2\right]+ \mathrm{Tr}\Big[ (\mathbb{I} - \mathbf{U}_m \mathbf{U}_m^\top) \Sigma_n \Big].
> > > > > > \end{align*}
> > > > > >
> > > > > >
> > > > > > As shown, $\mathcal{G}^{\prime}_2$ consists of three terms. The rightmost term depends both on the noise and the latent dimension ($m$) of the AE, as noted by the reviewer. The leftmost term is constant, and the middle (including the minus sign) and right terms always decrease with an increase of $m$. So overall, this definition of the test loss is monotonically decreasing as $m$ increases.
> > > > > >
> > > > > >
> > > > > > We hope our response clarifies the distinction we aim to highlight between noise memorization and signal learning as model size, training duration, and sample count vary.
> > > > > >
> > > > > >
> > > > > > [1] Johnstone, Iain M. "On the distribution of the largest eigenvalue in principal components analysis." The Annals of statistics 29.2 (2001): 295-327.
> > > > > >
> > > > > > [2] Donoho, David, and Matan Gavish. "Minimax risk of matrix denoising by singular value thresholding." (2014): 2413-2440.
> > > > > >
> > > > > > [3] Gavish, Matan, and David L. Donoho. "Optimal shrinkage of singular values." IEEE Transactions on Information Theory 63.4 (2017): 2137-2152.
> > > > > >
> > > > > > [4] Butsch, Lucas, and Vicky Fasen-Hartmann. "Estimation of the number of principal components in high-dimensional multivariate extremes." arXiv preprint arXiv:2505.22437 (2025).

---

> > > > > > > ### Comment · Reviewer_p8vq · 2025-08-14
> > > > > > >
> > > > > > > Thanks for your answer, I believe I now agree with the proof and the result!

---

### Comment · Action_Editor_gWtc · 2025-08-07
**Let's finish discussion and resolve all concerns regarding proof**

Dear Authors and Reviewers,

As you can see there is still ongoing discussion regarding correctness of the proof and main results (see discussion between Authors and Reviewer p8vq). Let's have a bit more time to finalize this discussion. I am ok extending discussion by 1 week (maybe 2 if I see the need) from today before going to recommendation phase.

Thanks,

AE.

---

### Comment · Action_Editor_gWtc · 2025-08-21

Dear Authors and Reviewers,

As the issue with the proof seems to be resolved, do you have anything else to discuss and resolve? If not and we have the final revision uploaded (Authors, please correct if you still plan to update the paper), then Reviewers, please submit your recommendation for the paper.

Thanks all,

AE.

---

> ### Author Response · Authors · 2025-08-22
>
> Dear Action Editor,
>
> Thank you for your message. We completed the revisions some time ago and, to the best of our understanding, we are currently awaiting the final decision. However, Reviewer RR8M mentioned that they are unable to see our response, even though we uploaded a detailed and thorough reply on June 21st, addressing all of their suggestions and revised the paper accordingly.
>
> We kindly ask you to look into this matter and ensure that our response is visible to the reviewers. To help expedite the process, we have reposted our original response from June 21st as a new comment to ensure that Reviewer RR8M can access it.
>
> We would greatly appreciate it if the process could be concluded without further delays, and we sincerely thank you for your support throughout.

---

> > ### Comment · Action_Editor_gWtc · 2025-08-22
> > **Request to Reviewer RR8M**
> >
> > Dear Reviewer RR8M,
> >
> > Could you confirm if you can access the latest revision and see all authors comments and responses?
> >
> > Thanks,
> >
> > AE

---

> > > ### Author Response · Authors · 2025-08-22
> > > **Response by the authos to the AE**
> > >
> > > Dear AE,
> > > As we noted earlier, Reviewer RR8M can not see our original response. It is indicated in the visibility option next to our response.
> > >
> > > You should be able to see it, and the system clearly indicates that our response is not visible to the reviewers. We can not edit the visibility option for that response. Could you please check if you can edit the visibility option?
> > > We would also be happy to know why the visibility of our response has changed (sometime after Jun 21st).
> > > Thanks,
> > > The authors

---

> > > > ### Comment · Action_Editor_gWtc · 2025-08-22
> > > >
> > > > I don't think anyone can change the visibility. So please repost all comments which need to be visible to reviewers again with controlling visibility.
> > > >
> > > > AE.

---

> > > ### Comment · Reviewer_RR8M · 2025-08-22
> > >
> > > Yes, I can now see the authors' response to my review.

---

### Comment · Action_Editor_gWtc · 2025-08-22
**Repost comments to Reviewer CF33 too**

Dear Authors,

The issue with your reply to Reviewers CF33 and RR8M was due to posting comments before the discussion phase started (before all three reviews were submitted): in this case the viewers of the posts are only AEs. Please check all your responses on to whom it is visible. So your latest post to Reviewer RR8M is not visible to everyone, however you need to post your response to Reviewer CF33 too as it is not visible to everyone.

Thanks in advance,

AE.

---

### Decision · Action_Editor_gWtc · 2025-09-04

**Recommendation:** Accept with minor revision

**Additional Comments:**

Please finalize the revision to have the final version within the TMLR guidelines and including latest clarifications / additions to the proof.

**Audience:**

Yes

**Audience Explanation:**

As pointed out by all reviewers, the paper considers double descent phenomenon in the context of SSL, specifically for autoencoders, and shows new contradictory to prior work results. As double descent is a pretty broad topic, the paper's results should be interesting to the community to understand the phenomenon better in broad conditions.

**Claims And Evidence:**

Yes

**Claims Explanation:**

Double descent is an interesting phenomenon which the community actively investigated in past years. The research mainly was focused on supervised models. One of the prior works claimed that it does not occur in SSL regime with autoencoders. In this paper, Authors in contrast show that double descent doesn’t occur in SSL with linear autoencoders both theoretically and empirically, while show empirically that double descent does occur in SSL with non-linear autoencoders across latent dimension, model size and data type.
Authors further show the application side of these observations: overparameterizing models to the double descent regime can improve downstream tasks (domain adaptation and anomaly detection).

**Strengths**

All reviewers pointed out important problems Authors investigated and important results they obtained: double descent happens at least in SSL with non-linear AEs contradicting prior work which claimed opposite for AEs. All reviewers also agreed that the paper provides comprehensive theoretical and empirical results to support the claims that double descent doesn’t occur for linear AE but does occur for non-linear AE. Also with minor comments all reviewers agreed that the paper is clearly and well written.

Some quotes from Reviewers:
- “Double descent in unsupervised settings remains a wide open question, and it is refreshingly interesting to see a paper that tackles this topic.”
- “The paper presents results that challenge the popular belief that double descent is specific to supervised learning.”
- “to the best of my knowledge the authors are right to claim that "this is the first demonstration of model and epoch, and sample-wise double descent in a fully unsupervised setting". This is an important contribution!”

**Weaknesses**

- Two Reviewers (CF33, RR8M) mainly suggested improving paper readability and formatting changes, while finding that paper is easy to follow and overall is solid. Authors addressed the comments during discussion and improved the flow of the paper. Both reviewers are happy with the changes.
- Reviewer p8vq mainly pointed out the issue with the proof for the main result regarding the linear AE property. During discussion Authors provided more detailed derivations and clarifications, so in the end Reviewer p8vq agreed with the proof and its correctness and meaningfulness.

In the end, all Reviewers are happy with the final revision. Authors have improved formatting and flow, and mainly have clarified correctness of the main theorem.

To me, the key contribution of the paper is showing contradiction with the “Gilad Lerman and Tyler Maunu. Fast, robust and non-convex subspace recovery“ prior work which claimed that double descent doesn’t occur in the SSL regime with AE. Results on double descent are now aligned again with prior observations in the supervised regime. Also, Authors clearly show that the cases of linear AE and non-linear AE represent entirely different regimes for double descent. Thus, the paper gives a push to the community to understand better if there are any fundamental differences between supervised and SSL regimes.